# Sample-Efficient Diffusion-based Reinforcement Learning with Critic Guidance

**Shutong Ding**[1][*] **Zejia Zhong**[1][*] **Zhongyi Wang**[1] **Ke Hu**[1] **Bikang Pan**[1] **Jingya Wang**[1] **Ye Shi**[1][†]

## Abstract

Recent advances in reinforcement learning (RL) have achieved great successes by leveraging the multimodality and exploration capability of diffusion policies. Among these approaches, one representative branch focuses on the sampling-based policy optimization. This design enables better exploration capability of the diffusion model, particularly at the beginning of training, but suffer from low exploitation in Q-value information, resulting in a slow policy convergence. Another branch pays attention to gradient-based policy optimization, which sufficiently exploits the gradient of the Q function yet tends to collapse into a unimodal policy with low diversity. To address this issue, we propose CGPO, **C**ritic-**G**uided diffusion **P**olicy **O**ptimization, which effectively balances exploration and exploitation with the training-free guidance technique integrated into the denoising process of diffusion policy. Concretely, CGPO steers action generation toward high-value regions defined by the critic network and uses the guided actions as regression objectives. In this manner, CGPO reduces the time required to obtain high-quality actions and improves final performance with better balance between the exploration-exploitation tradeoff. We validate the effectiveness of CGPO on 5 MuJoCo locomotion tasks, and CGPO achieves state-of-the-art performance compared with existing diffusion-based RL methods. Notably, CGPO is the first success to incorporate diffusion policy into real-world RL, with its superior performance on Franka robot arm grasping tasks. Our official page is released at https://dingsht.tech/cgpo-webpage.

[*] [1]ShanghaiTech University. Correspondence to: Ye Shi <shiye@shanghaitech.edu.cn>.

*Proceedings of the 43rd International Conference on Machine Learning*, Seoul, South Korea. PMLR 306, 2026. Copyright 2026 by the author(s).

## 1. Introduction

Recently, diffusion models have emerged as a powerful class of generative models, demonstrating strong capability in modeling complex and multimodal distributions (Ho et al., 2020; Song et al., 2020b;a). Motivated by these advantages, diffusion-based reinforcement learning has attracted increasing attention, as diffusion policies can naturally capture multimodal action distributions and encourage diverse exploration (Yang et al., 2023; Psenka et al., 2023; Ding et al., 2024; Wang et al., 2024; Ding et al., 2025). Existing approaches can be broadly divided into two lines of work: sampling-based optimization and gradient-based optimization. In sampling-based methods, actions sampled from a diffusion policy are reweighted according to their estimated values or advantages (Ding et al., 2024; Ma et al., 2025). In gradient-based methods, the gradients of a Q-network are used to guide the diffusion policy toward high-quality actions by training its noise prediction network (Yang et al., 2023; Psenka et al., 2023). Although these methods often exhibit reasonably high sample efficiency, their optimization mechanisms still inherently suffer from the exploration-exploitation trade-off.

In parallel, diffusion policies have also been actively explored in real-world robotic control, particularly in the context of imitation learning (Chi et al., 2023; Wu et al., 2025). By learning from large collections of human demonstrations, these methods have shown strong performance and robustness in manipulation tasks. Nevertheless, imitation-learning-based diffusion policies suffer from two fundamental limitations. First, collecting high-quality demonstrations in real-world settings is expensive and labor-intensive. Second, imitation learning is inherently bounded by the quality of human experts and therefore cannot systematically exceed demonstrated performance. These limitations motivate diffusion-based RL methods that learn directly from real-world interaction and surpass human demonstrations, yet this remains largely underexplored in existing works.

To bridge this gap, we first systematically analyze the limitations of existing works on diffusion-based RL and propose CGPO (Critic-Guided Diffusion Policy Optimization), a diffusion-based reinforcement learning framework that enables efficient training in real-world robotic systems without demonstrations. Unlike prior methods that rely on sampling

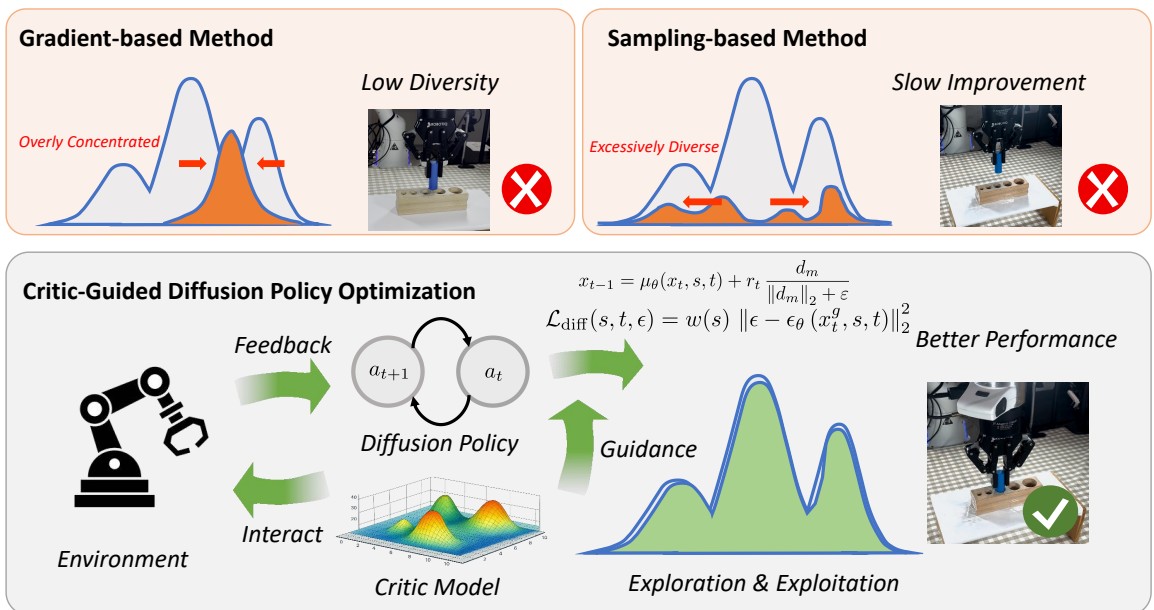

*Figure 1.* Overview of CGPO. Compared with Q-guided and sampling-based diffusion RL that suffers from low action diversity and slow improvement, CGPO performs critic-guided action generation during training to improve policy fitting and downstream task performance.

a large set of candidate actions followed by value-based reweighting, CGPO integrates classifier guidance into the diffusion denoising process, directly steering action generation toward high-value regions defined by a learned critic (Dhariwal & Nichol, 2021; Ho & Salimans, 2022). The resulting guided actions serve as optimization targets for updating the diffusion policy, eliminating the need for expensive candidate sampling while enabling more precise, gradient-informed policy improvement. By providing a continuous and directional optimization signal, CGPO effectively overcomes the refinement bottleneck of sampling diffusion RL methods. To summarize, our contributions are fourfold:

- We systematically analyze existing diffusion-based RL methods and interpret their limitations through the lens of exploration-exploitation trade-offs, providing a unified explanation for their suboptimal action discovery behavior and slow policy convergence.

- We propose CGPO, a novel diffusion-based reinforcement learning framework that integrates critic guidance directly into the denoising process of diffusion policies. By steering action generation toward high-value regions while preserving sufficient diversity, CGPO achieves a principled trade-off between exploration and exploitation.

- We also introduce several training recipe for critic-guided diffusion RL, including (i) a value-calibrated network that mitigates the impact of value drift on state-conditional weighting, and (ii) an overestimation-reduced critic target

construction to improve the reliability of the $Q$ signal used for both weighting and guidance.

- We conduct extensive evaluations in both simulation and real-world environments, including MuJoCo continuous control benchmarks and real-world manipulation tasks on a Franka robotic arm. The results show that CGPO consistently outperforms existing diffusion-based reinforcement learning methods as well as strong conventional baselines, demonstrating its effectiveness and practical relevance.

## 2. Related Work

**Diffusion Models in Reinforcement Learning.** Diffusion models have recently been applied to reinforcement learning across offline, off-policy, and on-policy settings. In offline reinforcement learning (Levine et al., 2020; Park et al., 2025), diffusion models are primarily used to policy networks to adhere to the behavioral cloning framework, enabling policy optimization by sampling actions or trajectories that remain within the data support. In off-policy reinforcement learning, DIPO (Yang et al., 2023) optimizes diffusion policies by augmenting a behavior-cloning objective with Q-function gradients, and relies on the diffusion models inherent stochasticity for exploration rather than explicitly regulating exploratory behavior. Notably, its extensions to multi-modal policy learning do not resolve this lack of explicit exploration control. QSM (Psenka et al., 2023) avoids backpropagation through the diffusion chain by matching the policy score to $\nabla_a Q(s, a)$, but disregards policy entropy and therefore requires heuristic noise injec-

tion for exploration. DACER (Wang et al., 2024) backpropagates gradients through the diffusion process but optimizes only expected Q-values and does not model a backward process; exploration is induced via additional Gaussian noise with entropy estimated approximately using a Gaussian mixture model. Similarly, QVPO (Ding et al., 2024) reweights diffusion losses using transformed Q-values. Soft Diffusion Actor-Critic (SDAC) (Ma et al., 2025) demonstrates improved sample efficiency but still requires careful energy-based modeling and incurs computational costs. In on-policy reinforcement learning, DPPO (Ren et al., 2024) proposes practical recipes for fine-tuning expressive diffusion policies using policy-gradient updates, enabling structured on-manifold exploration and stable long-horizon training. FPO (McAllister et al., 2025) estimates policy importance ratios via an ELBO objective, providing a scalable approximation but inducing asymmetric estimation bias. The estimates tend to be more reliable when the importance ratio increases than when it decreases, which can amplify variance and compromise training stability. GenPO (Ding et al., 2025) leverages exact diffusion inversion to construct invertible action mappings and introduces a doubled dummy action mechanism that achieves invertibility through alternating updates, thereby yielding tractable log-likelihoods.

**Diffusion Policy in Real-World Tasks.** Diffusion Policy (DP) (Chi et al., 2023) establishes diffusion models as a strong visuomotor policy class for real-world manipulation by generating short-horizon action sequences, or action chunks, conditioned on visual observations and robot proprioception, and training the denoiser via behavior cloning on demonstrations. Building on DP, subsequent work improves generalization by enriching the policy input and representation, for example DP3 (Ze et al., 2024) leverages simple 3D scene representations to provide geometry-aware conditioning for diffusion-based visuomotor control. Another line of work scales diffusion policies through large-scale pretraining, for example RDT-1B (Liu et al.) pretrains a large diffusion-transformer policy on broad multi-task robot experience and then adapts it to downstream tasks. A remaining challenge is how to use reinforcement learning to improve diffusion policies beyond demonstration-limited behavior. Although DSRL (Wagenmaker et al., 2025) introduces RL signals, it mainly fine-tunes the noise distribution (or sampling behavior) around a demonstration-anchored diffusion model, which still limits how far the policy can move away from demonstrations.

Despite the progress made by prior work, existing approaches either rely on diffusion models primarily for imitation learning or struggle to achieve strong exploration and efficient utilization in reinforcement learning settings. In contrast, our method incorporate a critic guidance into the denoising process of diffusion: it preserves the expressive generative representations of diffusion policies while leveraging value-based feedback and resulting in effective policy improvement.

# 3. Preliminaries

## 3.1. Reinforcement Learning

Reinforcement learning (RL) studies sequential decision-making under uncertainty, where an agent interacts with an environment to maximize long-term return. A discounted Markov decision process (MDP) is defined as $\mathcal{M} = (\mathcal{S}, \mathcal{A}, P, r, \gamma)$, where $\mathcal{S}$ and $\mathcal{A} \subseteq \mathbb{R}^d$ denote the state and continuous action spaces, $P(\cdot \mid s, a)$ is the transition kernel, $r : \mathcal{S} \times \mathcal{A} \to \mathbb{R}$ is the reward function, and $\gamma \in (0, 1)$ is the discount factor. A stochastic policy $\pi(a \mid s)$ specifies the agent's action distribution given state $s$. The standard RL objective is to maximize the expected discounted return:

$$J(\pi) = \mathbb{E}_{\pi, P}\Big[ \sum_{t=0}^{\infty} \gamma^t\, r(s_t, a_t) \Big]. \tag{1}$$

To evaluate a policy, the action-value function $Q^\pi(s, a)$ is defined as the expected return after taking action $a$ in state $s$ and following policy $\pi$ thereafter:

$$Q^\pi(s, a) := \mathbb{E}_{\pi, P}\Big[ \sum_{t=0}^{\infty} \gamma^t\, r(s_t, a_t) \,\Big|\, s_0 = s,\, a_0 = a \Big]. \tag{2}$$

The action-value function satisfies the Bellman expectation equation, which decomposes the return into the immediate reward and the discounted value of successor states:

$$Q^\pi(s, a) = r(s, a) + \gamma\, \mathbb{E}_{s' \sim P(\cdot \mid s, a),\, a' \sim \pi(\cdot \mid s')}\big[ Q^\pi(s', a') \big]. \tag{3}$$

In off-policy RL, transitions $(s, a, r, s')$ are collected by a behavior policy and stored in a replay buffer $\mathcal{D}$. Actor–critic methods learn a parameterized critic $Q_\omega(s, a)$ from Bellman-style regression targets and update a parameterized policy using critic-based policy improvement signals.

## 3.2. Diffusion Policies

A diffusion policy models the action distribution $\pi_\theta(a \mid s)$ as a conditional DDPM in action space, where the clean variable $x_0 \in \mathbb{R}^d$ corresponds to the action $a$ and the state $s$ serves as the conditioning input (Chi et al., 2023; Ho et al., 2020). Let $T$ be the number of diffusion steps and $\{\beta_t\}_{t=1}^T$ be a variance schedule, with $\alpha_t = 1 - \beta_t$ and $\bar\alpha_t = \prod_{i=1}^t \alpha_i$.

**Noising and denoising.** The forward process gradually corrupts $x_0$ into Gaussian noise, and the reverse process is parameterized by a denoiser $\epsilon_\theta(x_t, s, t)$ that predicts the injected noise at each step. Under the standard noise-prediction parameterization, the forward marginal and the

induced clean-action prediction are

$$x_t = \sqrt{\bar{\alpha}_t}\, x_0 + \sqrt{1 - \bar{\alpha}_t}\, \epsilon, \qquad \epsilon \sim \mathcal{N}(0, I),$$
$$\hat{x}_0(x_t, s, t) = \frac{1}{\sqrt{\bar{\alpha}_t}} \left( x_t - \sqrt{1 - \bar{\alpha}_t}\, \epsilon_\theta(x_t, s, t) \right).$$
$$(4)$$

Sampling starts from $x_T \sim \mathcal{N}(0, I)$ and applies the learned reverse transitions for $t = T, \ldots, 1$, returning the final action $a = x_0$ (or equivalently $\hat{x}_0$ at the last step).

**Training.** Given state–action pairs $(s, x_0)$, the denoiser is trained by the standard noise-prediction objective, which draws a random timestep and regresses the predicted noise to the injected noise:

$$\mathbb{E}_{s,\, x_0,\, t,\, \epsilon} \left[ \| \epsilon - \epsilon_\theta(x_t, s, t) \|_2^2 \right], t \sim \mathrm{Uniform}\{1, \ldots, T\}.$$
$$(5)$$

This objective yields a conditional generative model whose sampling-time trajectory can later be modified by external objectives (e.g., value guidance) without changing the diffusion parameterization.

### 3.3. Training-free Guidance

Training-free guidance steers the reverse diffusion trajectory using gradients of an external differentiable objective, without updating diffusion parameters or training an additional time-conditioned guidance model (Dhariwal & Nichol, 2021; Ho & Salimans, 2022; Chung et al., 2023; Song et al., 2023; Yu et al., 2023). Let $\ell(\cdot; y)$ be an objective defined on the clean prediction $\hat{x}_0$, where $y$ denotes the desired condition.

**Generic loss-guided update.** Given the mapping $x_t \mapsto \hat{x}_0(x_t, t)$ induced by the diffusion parameterization, guidance forms

$$g_t := \nabla_{x_t} \ell(\hat{x}_0(x_t, t); y). \tag{6}$$

A common plug-in correction modifies one reverse step by shifting the mean along $-g_t$:

$$x_{t-1} = \mu_\theta(x_t, t) - \eta_t\, g_t + \sigma_t\, \epsilon_t; \ \epsilon_t \sim \mathcal{N}(0, I), \tag{7}$$

where $(\mu_\theta, \sigma_t)$ follow the chosen reverse parameterization and $\eta_t > 0$ is a stepsize (Chung et al., 2023; Song et al., 2023).

**Other training-free variants.** Beyond the additive mean-shift in Equation (7), related training-free rules include score/noise mixing (e.g., classifier(-free) guidance) that alters the effective score used in the reverse step (Dhariwal & Nichol, 2021; Ho & Salimans, 2022), and projection/proximal updates that enforce constraints by projecting guided proposals back to a feasible set (He et al., 2024).

## 4. Methods

In this section, we present Critic-Guided Diffusion Policy Optimization (CGPO). Our motivation is that sampling-based diffusion-policy improvement becomes less effective as training proceeds: under a fixed sampling budget, action candidates drawn from a progressively concentrated policy tend to be similar, making critic scores less discriminative and weakening the practical update signal (Ding et al., 2024).

CGPO replaces multi-candidate selection with a single guided target per state. Specifically, we generate $a^g(s)$ by training-free loss-guided reverse diffusion under an action-value critic (Song et al., 2023; Yu et al., 2023), apply DSG only in the last $G$ denoising steps to focus guidance where clean predictions are more reliable (Yang et al., 2024), and train the actor with reweighted denoising regression on these guided targets (Ding et al., 2024). To keep the training signal well-conditioned, CGPO additionally (i) learns a lightweight base value network to stabilize the scale of critic-derived weights, and (ii) adopts overestimation-aware critic targets and aggregation so that the $Q$ signal used for weighting and guidance remains reliable throughout training (Van Hasselt et al., 2016; Kuznetsov et al., 2020).

### 4.1. Limitation in Sampling-based Optimization

Many online diffusion-policy RL methods improve the actor by drawing a finite set of candidate actions from the current policy and then using a learned critic to choose among these candidates (Ding et al., 2024). For each state $s$, the improvement step starts from a finite candidate set

$$a_i \sim \pi_\theta(\cdot \mid s), \qquad \mathcal{A}_K(s) = \{a_i\}_{i=1}^K. \tag{8}$$

The update signal is therefore sampling-driven: it is limited by the diversity of $\mathcal{A}_K(s)$ in Equation (8).

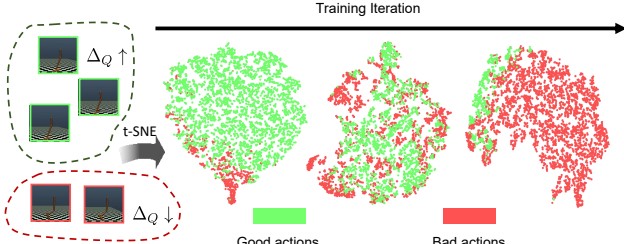

*Figure 2.* t-SNE visualization of sampled candidate actions on HalfCheetah-v3 at different training stages (left to right). Points are colored by critic-based labels (good vs. bad). As the policy concentrates, candidate diversity shrinks, and within-set separability decreases, consistent with a reduced critic contrast under finite candidate sampling.

A key limitation is that, as training proceeds, $\pi_\theta(\cdot \mid s)$ becomes more concentrated, so the candidates in Equation (8)

cover a narrower neighborhood under a fixed budget $K$. Consequently, the critic values $\{\hat{Q}_\psi(s, a_i)\}_{i=1}^K$ become less discriminative within $\mathcal{A}_K(s)$. A simple evaluation metric for this problem is the within-set critic contrast $\Delta_Q(s)$, which is defined as (9). This metric shrinks as candidates become similar; Figure 2 provides a toy example consistent with this effect. When $\Delta_Q(s)$ is small, selecting the best candidate yields only a marginal gain over a typical sample. Although we can alleviate it by increasing $K$ or the number of diffusion steps, the increased computation and control latency are undesirable for real-robot learning.

$$\Delta_Q(s) = \max_{a \in \mathcal{A}_K(s)} \hat{Q}_\psi(s, a) - \frac{1}{K} \sum_{i=1}^{K} \hat{Q}_\psi(s, a_i), \quad (9)$$

### 4.2. Critic-Guided Actions for Policy Improvement

Sampling-based improvement with a finite candidate set can weaken as training progresses(Section 4.1), motivating a more direct way to search for better actions. In conditional generation, classifier guidance steers diffusion trajectories by following gradients of a conditioning objective, often yielding substantially stronger controllability than pure sampling (Dhariwal & Nichol, 2021).This suggests a natural idea for diffusion-policy RL: use gradient-based guidance to synthesize higher-value actions and then train the policy toward these improved targets.

Classic classifier guidance requires a time-conditioned guidance model; in RL this would amount to learning a critic such as $\hat{Q}(s, x_t, t)$ that is accurate across diffusion noise levels, increasing training cost and introducing an undesirable bilevel coupling in online/robot learning. Training-free guidance instead backpropagates an objective on $\hat{x}_0$ through $x_t \mapsto \hat{x}_0(x_t, s, t)$ without an extra network (Chung et al., 2023; Song et al., 2023; Yu et al., 2023), but naive guidance can suffer from manifold deviation and thus needs conservative steps for stability (Yang et al., 2024; He et al., 2024).

Motivated by DSG (Yang et al., 2024), CGPO adopts a typicality-preserving constraint for guided reverse diffusion steps, which incorporate guidance while remaining close to the high-probability region of the unconditional reverse transition.

CGPO instantiates training-free guidance using a standard action-value critic on clean actions. For each replay state $s$ in an actor update, CGPO synthesizes a single refined target action $a^g(s)$ by running a guided reverse diffusion chain. The guidance signal is defined by the gradient of the critic objective with respect to the predicted clean action $\hat{x}_0(x_t, s, t)$:

$$\mathcal{L}_{\text{guide}}(x_t; s) = -\hat{Q}(s, \hat{x}_0(x_t, s, t)),$$
$$g_t = \nabla_{x_t} \mathcal{L}_{\text{guide}}(x_t; s), \quad (10)$$

where $\hat{Q}$ denotes a conservative scalar value estimate, the proof can be referred to **Appendix** A.5.

Standard unconditional reverse steps, $x_{t-1} = \mu_\theta(x_t, s, t) + \sigma_t \epsilon_t$, rely on the property that Gaussian perturbations in $d$ dimensions concentrate around a spherical shell with a typical radius $r_t = \sqrt{d}\, \sigma_t$. To incorporate guidance without violating this manifold structure, CGPO employs DSG. First, it maps the gradient $g_t$ to a constrained descent direction $d^\star$ located on this sphere:

$$d^\star = -r_t \frac{g_t}{\|g_t\|_2 + \varepsilon}, \quad (11)$$

where $\varepsilon$ is a constant for numerical stability. To preserve the stochastic nature of diffusion, DSG does not use $d^\star$ directly. Instead, it mixes this deterministic direction with the standard Gaussian noise $\epsilon_t \sim \mathcal{N}(0, I)$ using a guidance rate $\rho \in [0, 1]$:

$$d_m = \sigma_t \epsilon_t + \rho \left( d^\star - \sigma_t \epsilon_t \right). \quad (12)$$

Finally, the mixed direction $d_m$ is projected back onto the typical-radius shell to ensure the update magnitude matches the diffusion noise schedule, yielding the final update step:

$$x_{t-1} = \mu_\theta(x_t, s, t) + r_t \frac{d_m}{\|d_m\|_2 + \varepsilon}. \quad (13)$$

A key difference from conditional generation is that, in RL, the critic can be highly inaccurate and noisy early in training. Consequently, using raw critic gradients for guidance can be misleading and may destabilize refinement, especially when $\hat{x}_0(x_t, s, t)$ is still far from a plausible action at large noise levels.

To stabilize guidance, CGPO uses a conservative value estimate based on truncated quantiles (Kuznetsov et al., 2020). Specifically, with a distributional critic that outputs $M$ quantile estimates per critic in an $N$-member ensemble, let $Z_{(1)}^{(n)}(s, a) \leq \cdots \leq Z_{(M)}^{(n)}(s, a)$ denote the sorted quantiles. CGPO discards the top $k$ quantiles and aggregates the remainder as

$$\hat{Q}(s, a) = \frac{1}{N} \sum_{n=1}^{N} \left( \frac{1}{M - k} \sum_{m=1}^{M-k} Z_{(m)}^{(n)}(s, a) \right), \quad (14)$$

which yields a more conservative and stable guidance signal.In Equation (10), $\hat{Q}$ is instantiated by Equation (14), improving the reliability of the guidance direction.

In addition, CGPO applies DSG guidance only in the last $G$ denoising steps, while earlier steps follow the unconditional reverse transition. Restricting guidance to late steps reduces out-of-distribution critic queries and concentrates refinement in the regime where action-level changes become semantically meaningful. The guided sampler is used

only during training to produce $a^g(s)$; environment interaction and deployment use the unguided sampler to preserve runtime efficiency under real-robot control-loop latency constraints.

### 4.3. State Reweighting via Calibrated Value Signals

This section specifies the training objectives and update rules used in our implementation of CGPO. As discussed in Section 4.2, each actor update is supervised by a single training-time guided target $a^g(s)$, while environment interaction and deployment use the unguided sampler to preserve runtime efficiency.

Our implementation couples state reweighting with value-signal calibration to make critic-weighted diffusion updates reliable. Concretely, CGPO uses three tightly connected components: (i) weighted denoising regression on guided targets, together with a matched diffusion-entropy regularization (Ding et al., 2024); (ii) a value-calibrated network $V_\phi(s)$ that anchors the weight scale and reduces sensitivity to value drift; and (iii) an overestimation-aware critic target/aggregation scheme (DDQN targets (Van Hasselt et al., 2016) and truncated-quantile aggregation (Kuznetsov et al., 2020)) that improves the reliability of the $Q$ signal used by both weighting and guidance. We next describe these components in the same order.

**Value-calibrated network.**  A recurring issue in critic-weighted updates is that the scale of $\hat{Q}(s, a)$ can drift during training, making weights unstable and leading to overly aggressive or vanishing actor updates (Ding et al., 2024). CGPO introduces a lightweight value-calibrated network $V_\phi(s)$ (a small MLP) and forms weights from a rectified advantage:

$$w(s) = \max\left(\hat{Q}(s, a^g(s)) - V_\phi(s), 0\right). \tag{15}$$

Importantly, $V_\phi$ is trained to track the critic value of unguided actor samples, which anchors the value-calibrated network to the current policy distribution and avoids coupling it to the guided target generator. Concretely, let $\tilde{a}(s) \sim \pi_\theta(\cdot \mid s)$ be an unguided policy sample and define

$$y_V(s) = \mathrm{sg}\left(\bar{Q}(s, \tilde{a}(s))\right), \tag{16}$$

where $\mathrm{sg}(\cdot)$ stops gradients and $\bar{Q}$ uses the same critic aggregation rule as in Section 4.2. The base network is optimized by

$$\mathcal{L}_V = \mathbb{E}_{s \sim \mathcal{D}}\left[\left(V_\phi(s) - y_V(s)\right)^2\right]. \tag{17}$$

This value-calibrated network is used only to stabilize the weighting in the actor regression loss; it is not used in the guide objective. This design controls the *scale* of state reweighting, while the critic-side calibration in Section 4.2 improves the *reliability* of the underlying $Q$ signal.

**Weighted denoising and diversity regularization.**  Given guided targets $a^g(s)$, the diffusion actor is trained by weighted denoising regression, where $w(s)$ directly implements state reweighting in the actor loss:

$$\ell_{\text{diff}}(s, t, \epsilon) = w(s)\,\left\|\epsilon - \epsilon_\theta(x_t^g, s, t)\right\|_2^2,$$
$$\mathcal{L}_{\text{diff}} = \mathbb{E}_{s \sim \mathcal{D}, t, \epsilon}\left[\ell_{\text{diff}}(s, t, \epsilon)\right], \tag{18}$$
$$x_t^g = \sqrt{\bar{\alpha}_t}\, a^g(s) + \sqrt{1 - \bar{\alpha}_t}\, \epsilon,$$

where $t \sim \mathrm{Uniform}\{0, \ldots, T-1\}$ and $\epsilon \sim \mathcal{N}(0, I)$. To preserve action diversity, we additionally optimize a diffusion entropy regularization (Ding et al., 2024) using matched weighting:

$$\ell_{\text{ent}}(s, t, \epsilon) = w_{\text{ent}}(s)\,\left\|\epsilon - \epsilon_\theta(x_t^U, s, t)\right\|_2^2,$$
$$\mathcal{L}_{\text{ent}} = \mathbb{E}_{s \sim \mathcal{D}, a \sim U(\underline{a}, \bar{a}), t, \epsilon}\left[\ell_{\text{ent}}(s, t, \epsilon)\right], \tag{19}$$
$$x_t^U = \sqrt{\bar{\alpha}_t}\, a + \sqrt{1 - \bar{\alpha}_t}\, \epsilon,$$

where $w_{\text{ent}}(s) = \lambda_{\text{ent}}\, w(s)$ and $\lambda_{\text{ent}} > 0$ is a tunable hyperparameter.

**$Q$-signal calibration.**  Since the same $Q$ signal is used to assign weights in Equation (15) and provide guidance gradients for synthesizing $a^g(s)$, optimistic value targets can distort both the update emphasis and the direction of improvement. CGPO therefore adopts an overestimation-aware target construction and aggregation scheme. Specifically, critic targets follow a DDQN update, decoupling action selection from target evaluation (Van Hasselt et al., 2016). When using a distributional critic, we form the scalar $\hat{Q}$ by truncated-quantile aggregation, discarding the highest quantiles before computing targets (Kuznetsov et al., 2020). Together with the base value network $V_\phi$, this calibration yields a $Q$ signal that is stable in scale and reliable in ranking, which is crucial for critic-weighted diffusion updates.

**Overall procedure.**  The overall training procedure follows a standard off-policy actor–critic loop: a mini-batch of states is sampled from the replay buffer, guided targets $a^g(s)$ are synthesized for actor learning, the diffusion actor is updated by Equations (18) and (19), and the critic is updated by Bellman regression with the target construction above. Algorithm 1 summarizes the procedure. Overall, (iii) calibrates the $Q$ signal, (ii) anchors its scale for stable reweighting, and (i) turns the resulting state weights into effective diffusion-actor updates.

## 5. Experiments

We evaluate CGPO in simulation and on a real robot to assess both learning performance and practical deployability. Our experiments are designed to answer three questions: (i)

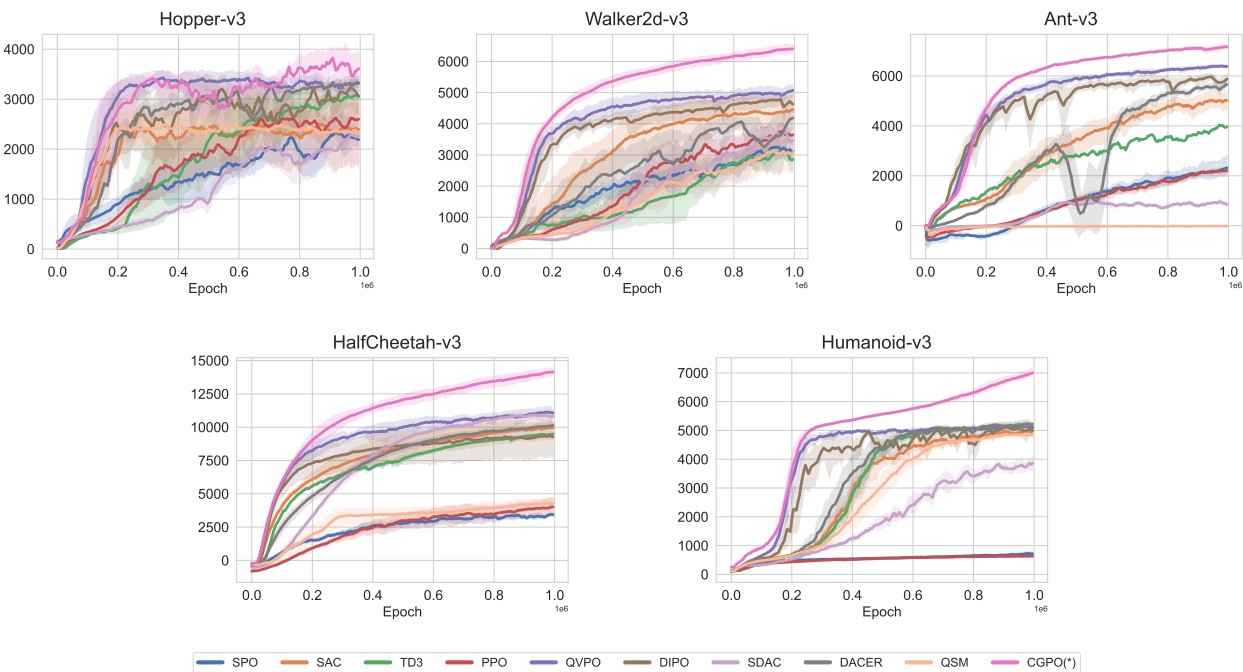

*Figure 3.* Learning curves on five MuJoCo v3 locomotion tasks over $10^6$ environment steps. Curves show the mean episodic return across five random seeds; shaded bands indicate $\pm 1$ standard deviation.

*Table 1.* Best episodic return achieved during training (mean $\pm$ standard deviation over five seeds). TD3 uses a deterministic unimodal actor; SAC/PPO/SPO use Gaussian stochastic policies; the remaining methods use diffusion policies. N/A indicates the method does not produce valid results under our setting.

| Task | | Unimodal | Gaussian | | | Diffusion | | | | | |
|---|---|---|---|---|---|---|---|---|---|---|---|
| | | TD3 | SAC | PPO | SPO | DIPO | QSM | DACER | QVPO | SDAC | CGPO(*) |
| **Ant-v3** | | 4583.8 (69.5) | 5030.9 (1000.3) | 2781.9 (74.1) | 2100.2 (302.4) | 6100.0 (177.3) | N/A | 5910.8 (82.3) | 6425.1 (67.6) | 1404.6 (105.3) | **7272.1 (74.6)** |
| **HalfCheetah-v3** | | 10388.6 (80.4) | 10616.9 (72.8) | 4773.5 (53.4) | 4008.2 (246.8) | 9555.0 (1654.3) | 3888.2 (632.6) | 10232.2 (272.5) | 11385.6 (164.5) | 11179.4 (325.1) | **14368.6 (310.6)** |
| **Hopper-v3** | | 3267.5 (8.5) | 2996.6 (111.9) | 3154.3 (426.2) | 2212.8 (988.4) | 3724.1 (240.7) | 2154.7 (998.2) | 3436.3 (33.2) | 3728.5 (13.8) | 2957.8 (323.7) | **4170.5 (165.6)** |
| **Humanoid-v3** | | 5353.5 (53.7) | 5159.7 (475.3) | 713.7 (85.9) | 797.4 (262.1) | 5280.1 (54.6) | 4793.1 (229.5) | 5291.3 (135.4) | 5306.6 (14.5) | 4429.6 (174.2) | **7131.4 (243.8)** |
| **Walker2d-v3** | | 3513.9 (40.7) | 4888.1 (80.0) | 3751.5 (609.1) | 3321.8 (1328.9) | 4917.1 (181.9) | 3613.4 (1443.5) | 4381.7 (226.2) | 5191.8 (60.2) | 4250.8 (411.9) | **6482.6 (170.9)** |

does CGPO improve online diffusion-policy reinforcement learning under a fixed interaction budget; (ii) does training-time critic-guided target synthesis yield more reliable policy improvement than sampling-based candidate selection when the policy distribution becomes concentrated; and (iii) can these gains be obtained without increasing deployment-time computation, by keeping rollout and evaluation unguided.

We first benchmark CGPO on five MuJoCo v3 locomotion tasks against strong model-free baselines and recent diffusion-policy methods, reporting learning curves and best achieved performance under identical training budgets. We then validate CGPO on a Franka Emika Panda system under a standardized intervention protocol, highlighting that the training-time guidance mechanism translates to robust real-world learning while respecting control-loop latency constraints.

### 5.1. Comparative Evaluation

We evaluate CGPO on five MuJoCo v3 locomotion benchmarks (Todorov et al., 2012) against representative online model-free RL baselines. The compared methods include off-policy algorithms (TD3 (Fujimoto et al., 2018), SAC (Haarnoja et al., 2018)), on-policy algorithms (PPO (Schulman et al., 2017), SPO (Chen et al., 2024)), and diffusion-policy / diffusion-actor–critic approaches (DIPO (Yang et al., 2023), DACER (Wang et al., 2024), QSM (Psenka et al., 2023), QVPO (Ding et al., 2024)), SDAC (Ma et al., 2025). All methods are trained for $10^6$ environment steps with the same interaction budget and evaluation protocol, and results are reported over five random seeds.

Figure 3 presents learning curves (mean return with $\pm$ one standard deviation across seeds). Table 1 reports, for each algorithm, the best episodic return attained during training, aggregated over the same five seeds; this metric summarizes

**Algorithm 1** Critic-Guided Diffusion Policy Optimization (CGPO)

---

**Input:** Diffusion policy $\pi_\theta$, critic $\hat{Q}$, replay buffer $\mathcal{D}$, diffusion steps $T$, guidance step $G$, learning rates $\eta_\theta, \eta_\psi$. Initialize actor parameters $\theta$ and critic parameters $\psi$.
**for** iteration $= 1, 2, \ldots$ **do**
    Collect transitions using the unguided sampler of $\pi_\theta$ and store $(s, a, r, s')$ in $\mathcal{D}$.
    Sample a mini-batch of states $\{s_i\}_{i=1}^B$ from $\mathcal{D}$.
    **for** each state $s$ in the mini-batch **do**
        Compute the refined target action $a^g(s)$ via critic-guided refinement
            (guidance objective in Equation (10); target synthesis procedure in Alg. 2).
    **end for**
    Compute $\mathcal{L}_{\text{diff}}$ and $\mathcal{L}_{\text{ent}}$ using Equations (18) and (19).
    Update actor: $\theta \leftarrow \theta - \eta_\theta \nabla_\theta (\mathcal{L}_{\text{diff}} + \alpha_{\text{ent}} \mathcal{L}_{\text{ent}})$.
    Update critic parameters $\psi$ by minimizing a conservative Bellman regression objective.
**end for**

---

the strongest performance reached within the fixed data budget.

Across all five tasks, CGPO achieves the best overall performance and exhibits consistently stable learning dynamics. Beyond final scores, the learning curves indicate that CGPO maintains effective optimization progress in regimes where sampling-based diffusion policy improvement tends to slow down. This behavior is consistent with CGPO's training-time critic-guided target generation: instead of relying on finite candidate sets to indirectly shape the actor update, CGPO constructs a single guided target per state and trains the actor by denoising regression on these targets. As a result, the policy update remains informative even when the diffusion policy becomes concentrated and the critic values of nearby samples become less separable.

Finally, we emphasize that critic guidance is used only during training to generate supervised targets for actor updates. Environment interaction and evaluation use the standard unguided sampler, keeping test-time inference cost and control-loop latency comparable to vanilla diffusion-policy execution, which is important for real-world deployment.

## 5.2. Ablation Study

To isolate the contribution of each component, we conduct ablations on Ant-v3, a representative high-dimensional locomotion task. As shown in Figure 5, we compare full CGPO with variants that remove critic guidance, replace DSG with naive guidance, remove DDQN, remove truncated-quantile aggregation, or remove the base value network.

**Guidance design.** Figure 5a shows that removing guidance substantially reduces performance, while naive guidance improves over the unguided variant but remains below CGPO. This indicates that critic gradients are useful, but the DSG-constrained update is important for injecting guidance in a way that remains compatible with the diffusion reverse process.

$Q$**-signal calibration.** Figures 5b and 5c show that removing either DDQN or truncated-quantile aggregation degrades learning. This supports the need for a reliable $Q$ signal, since the same critic is used for both guidance and actor weighting.

**Base value network.** As shown in Figure 5d, removing $V_\phi$ leads to lower and less stable performance. This confirms that anchoring the scale of critic-derived weights to unguided actor samples is important for stable state reweighting. Overall, these ablations show that CGPO benefits from both guided target synthesis and calibrated critic-weighted actor learning.

## 5.3. Real-world Experiments

**Tasks and Protocol.** We evaluate our method on two challenging tasks: Cube Stacking and Cylindrical Peg-in-Hole (Fig. 4). Following SiLRI (Zhao et al., 2025), we employ a four-stage intervention protocol that balances initial expert guidance with autonomous exploration. Corrective interventions are only triggered by recurring deviations, with a fallback to full demonstrations only after persistent failures. Further details about the robot setup can be found in the appendix.

**Quantitative Results.** As shown in Table 2, we compare our CGPO policy against the SAC baseline at 15,000 training steps. CGPO achieves a 80% success rate (16/20) in the cylindrical peg-in-hole task, outperforming SAC by 15%. Qualitatively, CGPO generates significantly smoother exploration trajectories, whereas the SAC baseline exhibits severe jittering during the insertion phase (Fig. 4, row 2).

*Table 2.* Success rates comparison of the cylindrical peg-in-hole task on the HIL-SERL Framework. We conducted 20 evaluation trials for each policy to calculate the success rate.

| Method | Success Rate (%) |
|---|---|
| HIL-SERL-SAC | 65 |
| HIL-SERL-CGPO (ours) | **80** |

## 6. Conclusion

In this work, we introduced CGPO, a new framework for diffusion-based reinforcement learning that addresses the

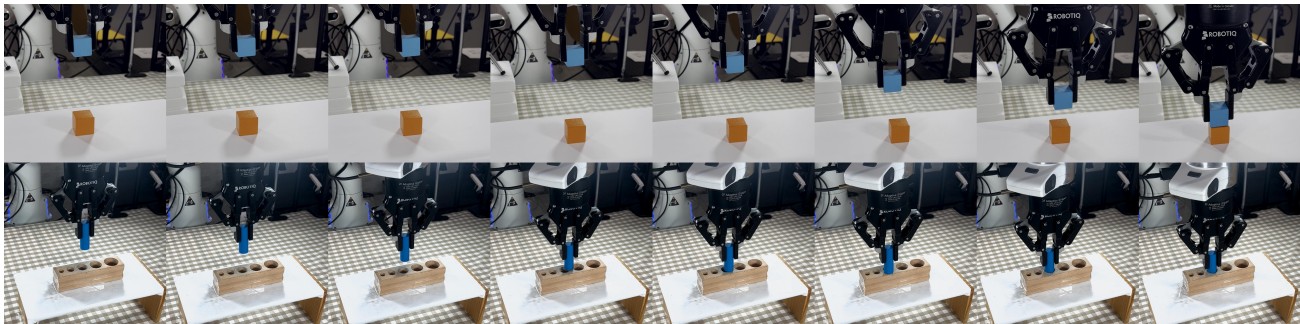

*Figure 4.* Sequential video frames of the real-world evaluation tasks. The top row shows the cube stacking task, and the bottom row shows the cylindrical peg-in-hole task.

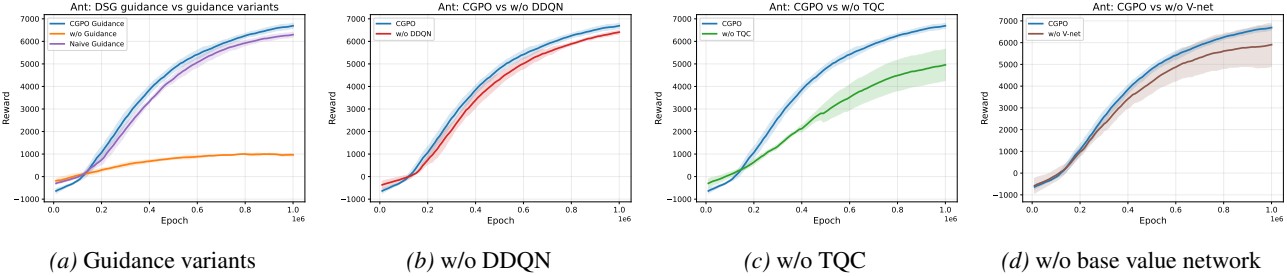

| (a) Guidance variants | (b) w/o DDQN | (c) w/o TQC | (d) w/o base value network |

*Figure 5.* Ablation study on Ant-v3. From left to right, we evaluate the effect of DSG guidance, DDQN target construction, truncated-quantile aggregation, and the base value network.

optimization limitations of existing weighted diffusion methods. By integrating critic guidance into the diffusion denoising process, CGPO generates high-quality action targets for diffusion policy improvement, avoiding extensive candidate sampling and enabling more precise policy refinement beyond early-stage learning. More importantly, CGPO demonstrates that diffusion-based reinforcement learning can be trained directly in real-world robotic systems from scratch, without reliance on large-scale human demonstrations. This work bridges the gap between diffusion policies and real-world reinforcement learning, and opens up a promising direction for combining generative models with critic-driven optimization in embodied intelligence. We believe CGPO provides a general and extensible foundation for future research on scalable, efficient, and real-worldready diffusion-based reinforcement learning.

## Acknowledgment

This work was supported by the National Natural Science Foundation of China (62303319, 62406195), HPC Platform of ShanghaiTech University, and MoE Key Laboratory of Intelligent Perception and Human-Machine Collaboration (ShanghaiTech University), Shanghai Engineering Research Center of Intelligent Vision and Imaging. This work was also supported in part by computational resources provided by Fcloud CO., LTD.

## Impact Statement

This work advances diffusion-based reinforcement learning by introducing CGPO, which utilize critic guidance that improves the explorationexploitation balance and enables diffusion policies to be trained directly through interaction. By reducing reliance on large-scale human demonstrations in imitation learning, CGPO has the potential to facilitate more practical and scalable deployment of diffusion policies in real-world robotic systems.

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

# A. Proofs

## A.1. Relative Entropy Policy Search

Relative Entropy Policy Search (REPS) derives a closed-form update for a new sampling distribution by maximizing expected return while constraining the KL divergence to a reference distribution:

$$\pi_{k+1} \in \arg\max_{\pi} J(\pi) \tag{20}$$

$$\text{s.t.} \quad \int_s d^{\pi_k}(s) \, \mathrm{KL}\big(\pi(\cdot \mid s) \,\|\, \pi_k(\cdot \mid s)\big) \, ds \le \varepsilon \tag{21}$$

where $J(\pi) = \sum_s d^{\pi_k}(s) \sum_a \pi(a \mid s) \, Q^{\pi_k}(s,a)$. This derivation begins by forming the Lagrangian of the constrained optimization problem presented above,

$$\mathcal{L}(\pi, \eta) = \int_s d^{\pi_k}(s) \int_a \pi(a \mid s) \, Q^{\pi_k}(s,a) - \eta \left( \int_s d^{\pi_k}(s) \int_a \pi(a \mid s) \log \frac{\pi(a \mid s)}{\pi_k(a \mid s)} - \varepsilon \right) \tag{22}$$

where $\beta$ is a Lagrange multiplier. Differentiating $\mathcal{L}(\pi, \eta)$ with respect to $\pi(a|s)$ and solving for the optimal policy results in the following expression for the optimal policy

$$\pi_{k+1}(a|s) = \frac{\pi_k(a \mid s) \exp\left( \frac{Q^{\pi_k}(s,a)}{\eta} \right)}{Z_\eta(s)}, \tag{23}$$

with $Z_\eta(s)$ being the partition function.

## A.2. Policy Evaluation

Given policy $\pi_k$, the update rule for the Q-function as:

$$(\mathcal{T}^{\pi_k} Q)(s,a) := r(s,a) + \gamma \mathbb{E}_{s' \sim P(\cdot|s,a), \, a' \sim \pi_k(\cdot|s')} \left[ Q(s', a') \right] \tag{24}$$

This formulation enables the application of standard convergence results for policy evaluation (Sutton et al., 1999).

## A.3. Policy Improvement

For the current value function $Q^k(s,a)$ and $\pi_k(a|s)$, it holds that:

$$\pi_{k+1}(a|s) = \arg\max_{\pi(\cdot|s)} \left\{ \int \pi(a|s) \, Q^{\pi_k}(s,a) \, da - \eta \, \mathrm{KL}\big(\pi(a|s) \,\|\, \pi_k(a|s)\big) \right\}. \tag{25}$$

There is:

$$\int \pi_{k+1}(a|s) \, Q^{\pi_k}(s,a) \, da - \eta \, \mathrm{KL}\big(\pi_{k+1}(a|s) \,\|\, \pi_k(a|s)\big) \ge \int \pi_k(a|s) \, Q^{\pi_k}(s,a) \, da - \eta \, \mathrm{KL}\big(\pi_k(a|s) \,\|\, \pi_k(a|s)\big) \tag{26}$$

$$= \int \pi_k(a|s) \, Q^{\pi_k}(s,a) \, da = V^{\pi_k}(s). \tag{27}$$

Since the KL term is non-negative, it follows that

$$\int \pi_{k+1}(a|s) \, Q^{\pi_k}(s,a) \, da \ge V^{\pi_k}(s). \tag{28}$$

Using the Bellman equation and applying this inequality recursively yields:

$$Q^{\pi_k}(s,a) = r_0 + \mathbb{E}[\gamma V^{\pi_k}(s)] \tag{29}$$

$$\le r_0 + \mathbb{E}[\gamma \mathbb{E}_{a_1 \sim \pi_{k+1}}(Q^{\pi_k}(s_1, a_1))] \tag{30}$$

$$= r_0 + \mathbb{E}\left[ \gamma r_1 + \gamma^2 \mathbb{E}_{a_2 \sim \pi_k(\cdot|s_2)} \left[ Q^{\pi_k}(s_2, a_2) \right] \right] \tag{31}$$

$$\le r_0 + \mathbb{E}\left[ \gamma r_1 + \gamma^2 \mathbb{E}_{a_2 \sim \pi_{k+1}(\cdot|s_2)} \left[ Q^{\pi_k}(s_2, a_2) \right] \right] \tag{32}$$

$$\cdots$$

$$= \mathbb{E}_{s \sim d^{\pi_{k+1}}, a \sim \pi_{k+1}} \left[ \sum_{t=0}^{\infty} \gamma^t r_t \right] = Q^{\pi_{k+1}}(s,a). \tag{33}$$

By unrolling the Bellman inequality, we obtain a monotonic improvement of the value function.

### A.4. Jensen Gap

Let x be a random variable with distribution $p(x)$. For a function $f$ that is convex or non-convex, the Jensen gap is defined as:

$$\mathcal{J}(f, x \sim p(x)) = \mathbb{E}[f(x)] - f(\mathbb{E}[x]) \tag{34}$$

If $f$ has L-smooth, then for $x, y \in \mathbb{R}$

$$f(x) \leq f(y) + \nabla f(y)^\top (x - y) + \frac{L}{2} \|x - y\|^2 \tag{35}$$

Set $y = \mathbb{E}[x]$:

$$f(x) \leq f(\mathbb{E}[x]) + \nabla f(\mathbb{E}[x])^\top (x - \mathbb{E}[x]) + \frac{L}{2} \|x - \mathbb{E}[x]\|^2 \tag{36}$$

$$\mathbb{E}[f(x)] \leq f(\mathbb{E}[x]) + \nabla f(\mathbb{E}[x])^\top \mathbb{E}[x - \mathbb{E}[x]] + \frac{L}{2} \mathbb{E}\|x - \mathbb{E}[x]\|^2 \tag{37}$$

$$\mathbb{E}[f(x)] \leq f(\mathbb{E}[x]) + \frac{L}{2} \mathbb{E}\|x - \mathbb{E}[x]\|^2 \tag{38}$$

$$\text{Jensen Gap} = \mathbb{E}[f(x)] - f(\mathbb{E}[x]) \leq \frac{L}{2} \mathbb{E}\|x - \mathbb{E}[x]\|^2 \leq \frac{L}{2} \text{tr}(\Sigma) \tag{39}$$

### A.5. Sampling Actions with Critic Guidance

Fixing $s$ and denote $Q(s, a) := \frac{1}{\eta} f(a) : \mathbb{R}^d \to \mathbb{R}$, and the target distribution is as followed:

$$\pi_0^*(a_0) := \frac{1}{Z_0} \pi_0(a_0) \exp(f(a_0)), \qquad Z_0 = \int \pi_0(a_0) \exp(f(a_0)) \, da_0. \tag{40}$$

The induced target marginal at noise level $t$ is

$$\pi_t^*(a_t) := \int q_t(a_t \mid a_0) \pi_0^*(a_0) \, da_0 = \frac{1}{Z_0} \int q_t(a_t \mid a_0) \pi_0(a_0) e^{f(a_0)} \, da_0 \tag{41}$$

$$= \frac{1}{Z_0} \pi_t(a_t) \int \frac{q_t(a_t \mid a_0) \pi_0(a_0)}{\pi_t(a_t)} e^{f(a_0)} \, da_0 \tag{42}$$

$$= \frac{1}{Z_0} \pi_t(a_t) \int \pi(a_0 | a_t) e^{f(a_0)} \, da_0 \tag{43}$$

$$= \frac{1}{Z_0} \pi_t(a_t) \mathbb{E}_{\pi(a_0|a_t)} \left[ e^{f(a_0)} \right]. \tag{44}$$

Hence, the guided score can be approximated as

$$\nabla_{a_t} \log \pi_t^*(a_t) = \nabla_{a_t} \log \pi_t(a_t) + \nabla_{a_t} \log \mathbb{E}_{\pi(a_0|a_t)} \left[ e^{f(a_0)} \right] \tag{45}$$

$$\approx \nabla_{a_t} \log \pi_t(a_t) + \nabla_{a_t} \mathbb{E}_{\pi(a_0|a_t)} [f(a_0)] \tag{46}$$

$$\approx \nabla_{a_t} \log \pi_t(a_t) + \nabla_{a_t} f(\mathbb{E}_{\pi(a_0|a_t)}[a_0]) \tag{47}$$

$$= s_\theta(a_t) + \nabla_{a_t} Q(s, \hat{a}_0(a_t)), \tag{48}$$

where $s_\theta(a_t)$ is the score of the diffusion model and $\hat{a}_0(a_t)$ is the posterior estimation via tweedie's formula.

## B. More Details on Practical Implementation

### B.1. Diffusion Guidance with Spherical Gaussian Algorithm

This appendix provides the detailed pseudocode for the value-guided refinement operator used by CGPO to synthesize a single refined target action $a^g(s)$ for each training state $s$. The refinement runs a reverse diffusion chain initialized from

Gaussian noise and injects a critic-based guidance signal only in the final $G$ denoising steps. This design keeps early denoising steps unguided (when $\hat{x}_0$ is still highly noisy and critic signals are less reliable) while concentrating refinement in late steps where action-level changes are semantically meaningful and sampling-based improvement is most prone to plateau.The guidance step follows DSG (Yang et al., 2024), which constrains the perturbation to the typical-radius spherical shell of the reverse Gaussian transition, mitigating manifold deviation and preserving stochasticity through a convex mixture with the unconditional Gaussian perturbation.

Algorithm 2 uses the standard DDPM $\epsilon$-parameterization to compute $\hat{x}_0(x_t, s, t)$ and the reverse mean $\mu_\theta(x_t, s, t)$. When $t > G$, the update reduces to the unconditional reverse transition. When $t \leq G$, the algorithm computes a guidance direction from the critic evaluated at the clean prediction $\hat{x}_0$ and applies the DSG constrained update. The refined target is returned as the final clean prediction produced by the reverse chain.

---

**Algorithm 2** Value-Guided Refinement with DSG

---

**Input:** State $s$, critic $\hat{Q}_\psi$, guidance step $G$, guidance rate $\rho$, noise scale $\sigma_t$.
**Output:** Refined action $a^g(s)$.
Initialize $x_T \sim \mathcal{N}(0, \mathbf{I})$.
**for** $t = T, T-1, \ldots, 1$ **do**
    $\hat{x}_0 \leftarrow \frac{1}{\sqrt{\bar{\alpha}_t}}\left(x_t - \sqrt{1 - \bar{\alpha}_t}\epsilon_\theta(x_t, s, t)\right)$
    $\mu_\theta(x_t, t) \leftarrow \frac{\sqrt{\alpha_t}(1 - \bar{\alpha}_{t-1})}{1 - \bar{\alpha}_t}x_t + \frac{\sqrt{\bar{\alpha}_{t-1}}\beta_t}{1 - \bar{\alpha}_t}\hat{x}_0$
    **if** $t > G$ **then**
        Sample $\epsilon_t \sim \mathcal{N}(0, \mathbf{I})$ and update $x_{t-1} \leftarrow \mu_\theta(x_t, t) + \sigma_t\epsilon_t$.
    **else**
        **Compute guidance gradient:** $g_t = \nabla_{x_t} - \hat{Q}(s, \hat{x}_0(x_t, s, t))$.
        **Determine constrained direction:**
          $d^\star \leftarrow -\sqrt{d}\,\sigma_t\frac{g_t}{\|g_t\|_2 + \varepsilon}$    {Steepest-descent on sphere}
        **Blend with Gaussian perturbation:**
        Sample $\epsilon_t \sim \mathcal{N}(0, \mathbf{I})$ and compute mixed direction:
        $d_m \leftarrow \sigma_t\epsilon_t + \rho\left(d^\star - \sigma_t\epsilon_t\right)$
        **Spherical projection and update:**
          $x_{t-1} \leftarrow \mu_\theta(x_t, t) + \sqrt{d}\,\sigma_t\frac{d_m}{\|d_m\|_2}$
    **end if**
**end for**
**return** $a^g(s) = \hat{x}_0$ at $t = 0$.

---

## B.2. Real-World RL System Settings and Training Details

We implemented the real-world RL system on a Franka Emika Panda arm with a Robotiq 2F-85 gripper, controlled via the Deoxys (Zhu et al., 2022) interface within the HIL-SERL (Luo et al., 2024) framework. Specifically, a 3Dconnexion SpaceMouse is integrated into the system to enable precise teleoperation. Within the HIL-SERL (Luo et al., 2024) framework, this interface allows the human operator to seamlessly switch between collecting initial expert demonstrations and applying corrective interventions to guide the agent. As shown in Figure 6, our visual setup integrates an eye-in-hand ZED Mini camera ($672 \times 376$ resolution) and a fixed third-person RealSense D435i ($640 \times 480$ resolution). The policy input (state space) combines these multi-view RGB images with proprioceptive statesspecifically, the end-effector Cartesian position, velocity, and gripper position. The policy outputs actions in the form of end-effector Cartesian position commands. The SAC and CGPO policy take approximately 1 hours to converge. Notably, the human intervention time for both methods is approximately 20 minutes.

## C. Hyper-parameters

### C.1. MuJoCo Gym

All experiments are conducted on a single NVIDIA GeForce RTX 4090D GPU (24GB) with an Intel(R) Core(TM) i9-14900K CPU. We implement SAC, TD3, PPO, SPO, and DIPO by building on their public reference codebases and keep the

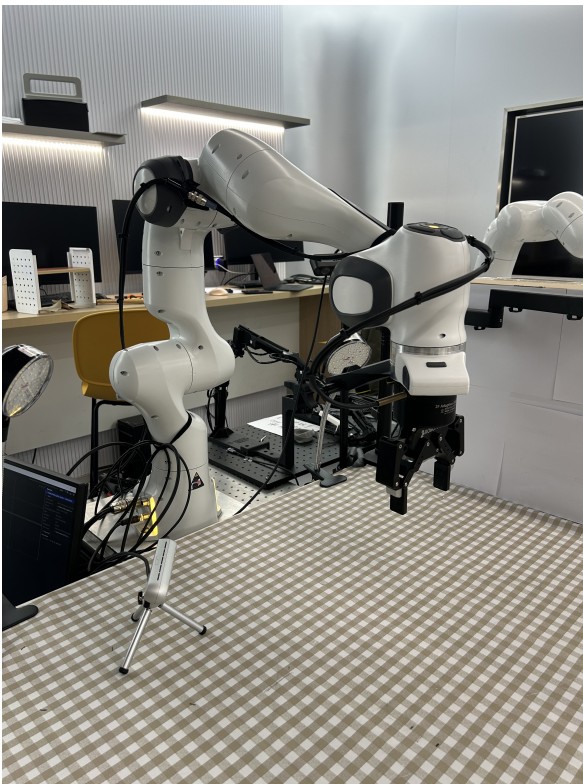

*Figure 6.* Experimental setup with the Franka Emika Panda robot and Robotiq 2F-85 gripper.

*Table 3.* Hyperparameters for the real-world experiments.

| Name | Value | Note |
|---|---|---|
| max_traj_length | 100 | Episode length limit for training |
| batch_size | 256 | Mini-batch size |
| cta_ratio | 2 | Collect-to-update ratio |
| discount | 0.98 | $\gamma$ |
| replay_buffer_capacity | 20000 | Replay buffer size |
| steps_per_update | 50 | Actor network parameters update steps |
| encoder_type | resnet10-pretrained | Visual encoder type |

overall training and evaluation protocol consistent across methods. The complete hyper-parameter configurations for all compared algorithms are summarized in Table 4, while CGPO-specific settings are reported separately in Table 5.

For CGPO, we intentionally avoid environment-by-environment tuning and use a single task-invariant configuration across all MuJoCo v3 tasks (Ant, HalfCheetah, Hopper, Humanoid, and Walker2d), including the sampling budget $N_e$, selection counts $(K_b, K_t)$, guidance parameters $(G, \rho)$, and entropy weight $\lambda_{\text{ent}}$ (Table 5), to reduce reproduction overhead and isolate algorithmic effects.

### C.2. Real-World Symtem

Hyperparameters used for the real-world experiments are shown in Table 3.

*Table 4.* Hyperparameters used in the experiments.

|  | CGPO | QVPO | DIPO | SAC | TD3 | SPO | PPO | DACER |
|---|---|---|---|---|---|---|---|---|
| **Network** | | | | | | | | |
| Hidden layers | 2 | 2 | 2 | 2 | 2 | 2 | 2 | 3 |
| Hidden width | 256 | 256 | 256 | 256 | 256 | 64 | 256 | 256 |
| Activation | mish | mish | mish | relu | relu | tanh | tanh | mish |
| **Optimization** | | | | | | | | |
| Batch size | 256 | 256 | 256 | 256 | 256 | 256 | 256 | 256 |
| Discount $\gamma$ | 0.99 | 0.99 | 0.99 | 0.99 | 0.99 | 0.99 | 0.99 | 0.99 |
| Target smoothing $\tau$ | 0.005 | 0.005 | 0.005 | 0.005 | 0.005 | 0.0 | 0.005 | 0.005 |
| Actor lr $\eta_\pi$ | $3 \times 10^{-4}$ | $3 \times 10^{-4}$ | $3 \times 10^{-4}$ | $3 \times 10^{-4}$ | $3 \times 10^{-4}$ | $3 \times 10^{-4}$ | $7 \times 10^{-4}$ | $1 \times 10^{-4}$ |
| Critic lr $\eta_Q$ | $3 \times 10^{-4}$ | $3 \times 10^{-4}$ | $3 \times 10^{-4}$ | $3 \times 10^{-4}$ | $3 \times 10^{-4}$ | $3 \times 10^{-4}$ | $7 \times 10^{-4}$ | $1 \times 10^{-4}$ |
| Grad norm clip | N/A | N/A | 2 | N/A | N/A | 0.5 | 0.5 | N/A |
| Replay buffer size | $10^6$ | $10^6$ | $10^6$ | $10^6$ | $10^6$ | $10^6$ | $10^6$ | $10^6$ |
| **Exploration / Regularization** | | | | | | | | |
| Entropy coef. | N/A | N/A | N/A | 0.2 | N/A | N/A | 0.01 | learned $\alpha$ |
| Value loss coef. | N/A | N/A | N/A | N/A | N/A | N/A | 0.5 | N/A |
| Exploration noise | N/A | N/A | N/A | N/A | $\mathcal{N}(0, 0.1)$ | N/A | N/A | $\mathcal{N}(0, (0.15\alpha)^2)$ |
| Policy noise | N/A | N/A | N/A | N/A | $\mathcal{N}(0, 0.2)$ | N/A | N/A | N/A |
| Noise clip | N/A | N/A | N/A | N/A | 0.5 | N/A | N/A | N/A |
| Use GAE | N/A | N/A | N/A | N/A | N/A | True | True | N/A |
| **Diffusion** | | | | | | | | |
| Diffusion steps $T$ | 20 | 20 | 20 | N/A | N/A | N/A | N/A | 20 |

## D. Additional Results

### D.1. Parameter Analysis

To further understand the role of training-time refinement in CGPO, we conduct a parameter analysis on two key hyperparameters of CGPO: the guidance rate $\rho$ and the guidance step $G$ (number of guided denoising steps). We report results on Ant-v3 as a representative example; similar trends are observed across other locomotion tasks.

**Effect of guidance rate $\rho$.** The left panel of Figure 7 compares different guidance rates ($\rho \in \{0.35, 0.65, 0.95\}$). We observe that stronger guidance generally leads to better asymptotic performance, with $\rho = 0.95$ achieving the highest final return, while smaller $\rho$ values converge to lower plateaus. Intuitively, increasing $\rho$ places more emphasis on the

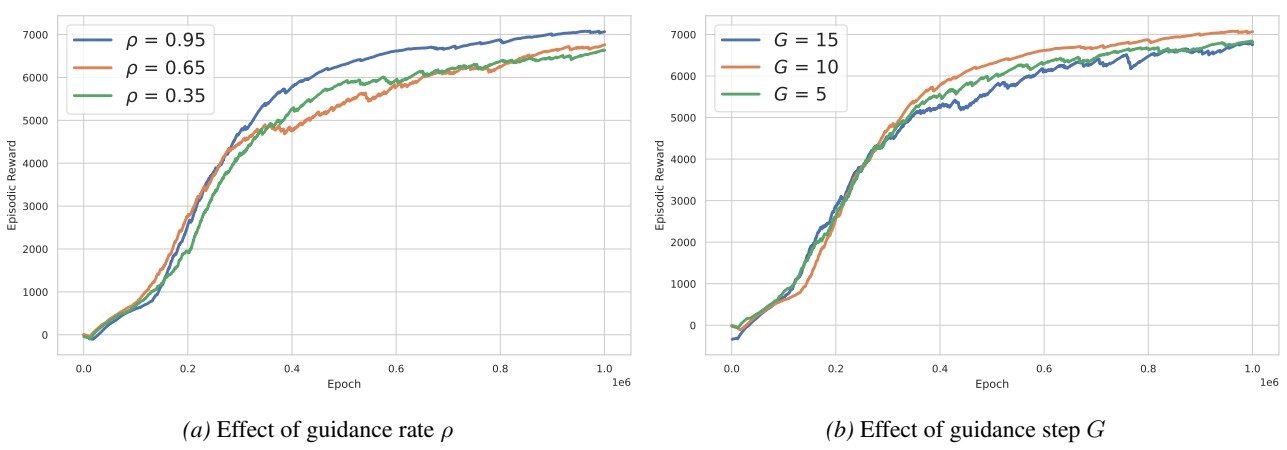

*(a)* Effect of guidance rate $\rho$        *(b)* Effect of guidance step $G$

*Figure 7.* Parameter analysis of CGPO on Ant-v3

*Table 5.* CGPO hyper-parameters on MuJoCo v3 tasks.

| | Ant-v3 | HalfCheetah-v3 | Hopper-v3 | Humanoid-v3 | Walker2d-v3 |
|---|---|---|---|---|---|
| **Sampling** | | | | | |
| # uniform samples $N_e$ from $\mathcal{U}(\underline{a}, \overline{a})$ | 10 | 10 | 10 | 10 | 10 |
| **Action selection** | | | | | |
| Behavior selection count $K_b$ | 4 | 4 | 4 | 4 | 4 |
| Target selection count $K_t$ | 4 | 4 | 4 | 4 | 4 |
| **Guidance** | | | | | |
| Guidance step $G$ | 10 | 10 | 10 | 10 | 10 |
| Guidance rate $\rho$ | 0.95 | 0.95 | 0.95 | 0.95 | 0.95 |
| **Regularization** | | | | | |
| Entropy weight $\lambda_{\mathrm{ent}}$ | 0.02 | 0.02 | 0.02 | 0.02 | 0.02 |

critic-induced refinement direction in DSG, thereby producing higher-quality refined targets for the subsequent denoising regression update. In practice, we find that a relatively large $\rho$ is beneficial as long as guidance is restricted to the late denoising regime.

**Effect of guidance step $G$.** The right panel of Figure 7 studies the number of guided denoising steps ($G \in \{5, 10, 15\}$) with the same total diffusion steps. A moderate step ($G = 10$) performs best, while using too few guided steps ($G = 5$) yields weaker gains, consistent with insufficient refinement strength. Conversely, pushing guidance too early ($G = 15$) can slightly degrade performance, which is consistent with the RL-specific challenge that early-step denoising variables (and their predicted clean actions) are more noise-dominated, making critic-based guidance less reliable and more prone to distribution shift. Overall, these results support the design choice of applying guidance only in the final denoising steps.

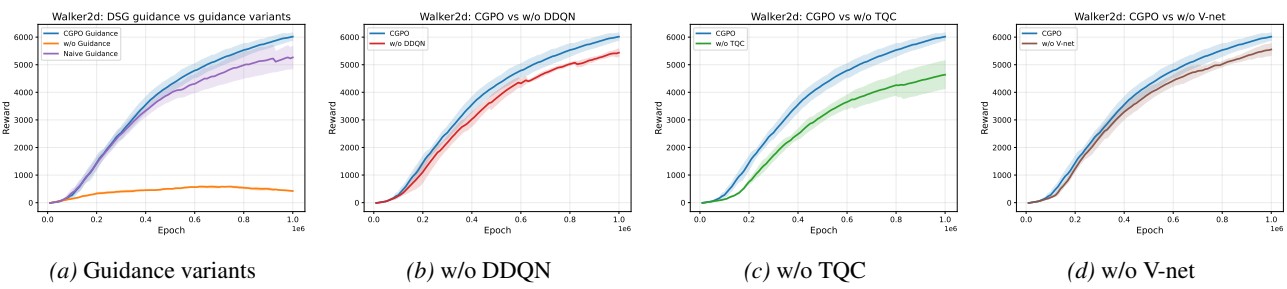

*(a)* Guidance variants      *(b)* w/o DDQN      *(c)* w/o TQC      *(d)* w/o V-net

*Figure 8.* Component ablations of CGPO on Walker2d-v3.

### D.2. Additional Ablation Study

To further isolate the contribution of each component, we conduct additional ablations on Walker2d-v3, as shown in Figure 8. The left panel compares DSG guidance with unguided target generation and naive guidance. Removing guidance leads to weaker learning, while naive guidance improves over the unguided variant but still underperforms full CGPO, indicating that the form of guidance is important. This supports the use of DSG as a structured guidance rule that is better aligned with the diffusion reverse process.

The middle two panels evaluate the effect of DDQN target construction and truncated-quantile aggregation. Removing either component degrades performance, suggesting that overestimation control and stable scalar $Q$ aggregation are important for both target generation and actor weighting. The right panel removes the base value network. The resulting performance drop confirms that anchoring the scale of critic-derived weights to unguided actor samples helps stabilize state reweighting. Overall, these results are consistent with the Ant-v3 analysis and verify that CGPO benefits from both guided target synthesis and calibrated critic-weighted actor learning.

### D.3. Quantitative Analysis of Critic Contrast

To further validate the mechanism discussed in Section 4.1, we analyze the evolution of the critic contrast $\Delta_Q(s)$ during training. Recall that in Equation (9), $\Delta_Q(s)$ measures the value gap between the best sampled candidate and the average value of the finite candidate set. A larger $\Delta_Q(s)$ indicates that the critic can clearly distinguish a better action from typical samples, whereas a smaller value suggests that sampling-based improvement provides only a weak update signal.

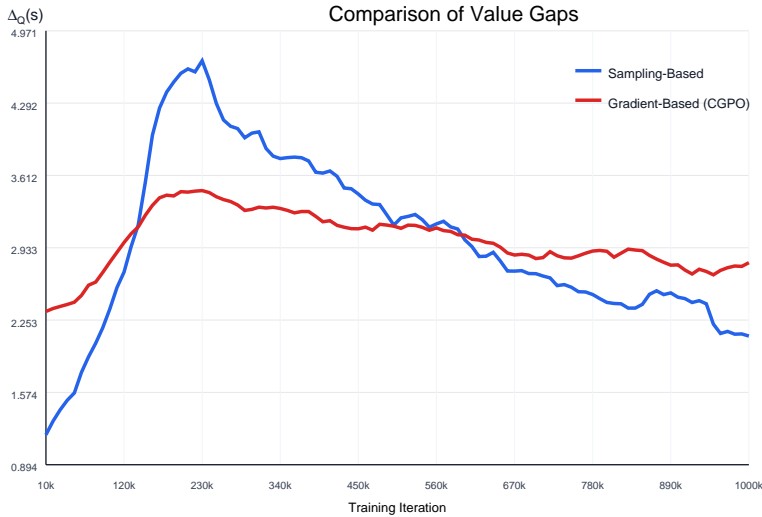

*Figure 9.* Evolution of critic contrast $\Delta_Q(s)$ on Ant-v3. For the sampling-based variant, we sample $K = 64$ candidate actions and compute $\Delta_Q(s)$ according to Equation (9). For CGPO, we compute the value gap induced by the DSG-guided action target. Sampling-based improvement exhibits a large value gap early in training but gradually loses contrast as training progresses, while CGPO maintains a more stable improvement signal.

As shown in Figure 9, the sampling-based method obtains a relatively large $\Delta_Q(s)$ in the early stage, indicating that finite candidate sampling can initially discover actions with noticeably higher critic values. However, as training proceeds and the policy distribution becomes more concentrated, the value gap decreases and becomes more fluctuating. This supports the analysis in Section 4.1: when sampled candidates become similar, the critic values within the candidate set become less separable, and the effective improvement signal weakens.

In contrast, CGPO variant maintains a more stable value gap in the later stage of training. This suggests that critic-guided target synthesis can provide a more persistent improvement signal after sampling-based candidate selection begins to lose discriminative power. Together with the learning-curve results, this quantitative analysis supports the central design motivation of CGPO: replacing finite candidate search with training-time critic-guided target synthesis can alleviate the weakening of sampling-driven policy improvement.

