# OpenReview forum: "Sample-Efficient Diffusion-based Reinforcement Learning with Critic Guidance"
_ICML.cc/2026/Conference — ICML 2026 regular_

### Official Review · Reviewer_8fdp · 2026-03-10

**Soundness:** 3
**Presentation:** 3
**Significance:** 3
**Originality:** 3
**Overall Recommendation:** 4
**Confidence:** 4

**Summary:**

This paper proposes CGPO (Critic-Guided Diffusion Policy Optimization), aiming to address the exploration-exploitation trade-off in diffusion-based RL by integrating training-free classifier guidance into the reverse denoising process. The core contribution is replacing the standard multi-candidate sampling-and-ranking approach with a single guided denoising trajectory that directly steers action generation toward high-value regions using critic gradients. Specifically, CGPO applies DSG guidance in the last few denoising steps, rotating noise perturbations toward critic-preferred directions while preserving their expected magnitude on the spherical shell. The resulting guided actions serve as weighted regression targets for the diffusion actor, with weights computed as rectified advantages via a value-calibrated baseline. Guidance is used at training time only, deployment runs unguided. Evaluated on five MuJoCo v3 tasks and real-world Franka Panda manipulation, CGPO outperforms existing diffusion-based RL methods and classical baselines in simulation, and achieves higher success rates than SAC on a real-robot peg-in-hole task.

**Compliance With Llm Reviewing Policy:**

Affirmed.

**Final Justification:**

Following the additional ablation studies, I remain a weak accept on this paper.

**Key Questions For Authors:**

In addition to the concerns raised in the weaknesses section:

1. In sparse-reward settings, the rectified advantage weights likely stay near zero until the agent first encounters reward, effectively disabling guidance. When rewards are finally observed, would the hard gating in max(Q̂ − V_ϕ, 0) create a sharp transition from inactive to active weights? And since the guided actions become supervised regression targets rather than soft policy gradient signals, would this commit the policy more strongly to potentially unreliable critic estimates? Has this scenario been tested, and does the policy overfit to early rewarding experiences?
2. What are the values of N (ensemble size), M (quantiles per critic), and k (truncation count)?

**Limitations:**

yes

**Strengths And Weaknesses:**

## Strengths
1. Well-motivated problem analysis. The paper clearly identifies why existing diffusion-based RL methods underperform: weighted methods lose their improvement signal as candidate actions become clustered (the $\Delta_Q$ contrast shrinks), while gradient methods sacrifice diversity. This provides a clean motivation for replacing multi-candidate selection with guided target synthesis.
2. Clean guidance mechanism. The DSG constraint, rotating noise direction toward critic-preferred regions while preserving magnitude on the spherical shell, is geometrically well-motivated and avoids the manifold deviation that naive mean-shift guidance would cause. Restricting guidance to the last few denoising steps where clean-action predictions are reliable is a sensible RL-specific adaptation.
3. Consistent improvements over diffusion baselines with task-invariant hyperparameters. CGPO shows substantial gains over QVPO across all five tasks with relatively small variance, using a single configuration throughout, suggesting the method is reasonably robust and not overfit to specific benchmarks.
4. Clean separation of training and deployment. Guidance is used only during training to produce regression targets; deployment uses the standard unguided sampler with identical latency to any diffusion policy.
5. Real-world validation. Demonstrating diffusion-policy RL trained from scratch on a physical Franka Panda arm is a meaningful practical contribution, especially the qualitative observation of smoother exploration trajectories compared to SAC.


## Weaknesses
1. Narrow ablation study. Only two hyperparameters (ρ, G) are varied on a single task (Ant-v3). At minimum, the paper should include CGPO without guidance to isolate whether the stabilization tricks alone close the gap. Other valuable ablations, like removing V_ϕ (using raw Q-values for weighting), removing truncated quantiles (using standard double Q), or comparing DSG guidance against naive mean-shift guidance, are also absent. Without these, we cannot attribute the improvement to the novel guidance mechanism versus the standard critic improvements.
2. Missing modern baselines. While it has become common in RL to benchmark against SAC and TD3 in their original formulations, this comparison is misleading here: CGPO uses substantially more parameters and per-step compute than the baselines it is compared against. Recent methods like SimBa [1], which achieves state-of-the-art performance by simply scaling network architecture, are absent. A compute-matched or parameter-matched comparison is needed to isolate the contribution of diffusion + guidance from the contribution of better networks and critics.
3. No sparse-reward simulation benchmarks. The paper's narrative emphasizes exploration-exploitation balance, yet all simulation experiments use dense-reward MuJoCo locomotion. The real-world peg-in-hole task is arguably semi-sparse (success/failure), which partially addresses this concern, but is insufficient as convincing evidence for the exploration claim.

Minor typos:
- three-fold -> four bullet points
- tradd-off -> trade-off
-  The claim of being "the first success to incorporate diffusion policy into real-world RL" needs qualification given DSRL[2] and other fine-tuning methods. -> from scratch, without demonstrations."

[1] SimBa: Simplicity Bias for Scaling Up Parameters in Deep Reinforcement Learning, Lee et al., https://openreview.net/forum?id=jXLiDKsuDo

[2] Steering Your Diffusion Policy with Latent Space Reinforcement Learning, https://arxiv.org/pdf/2506.15799

---

> ### Author Rebuttal · Authors · 2026-03-31
>
> We sincerely thank Reviewer 8fdp for the detailed and constructive comments, and provide a point-by-point response below.
>
> > **Q1**: Narrow ablation study ... we cannot attribute the improvement to the novel guidance mechanism versus the standard critic improvements.
>
>
> **A1**: We add more ablation studies at (https://anonymous.4open.science/r/icml-cgpo-C720). We divided the ablation into the following four groups: (1) Remove the $V_{\phi}$ network. (2) Remove the double q network (DDQN). (3) Do not perform guidance, perform naive guidance, and perform DSG guidance. (4) Replace the Truncated Quantiles Critic with the standard Critic. To avoid randomness in the experiment, we ran three seeds in each of the Ant-v3 and Walker2d-v3 environments.
>
>
> > **Q2**: Missing modern baselines ... Recent methods like SimBa [1], which achieves state-of-the-art performance by simply scaling network architecture, are absent. A compute-matched or parameter-matched comparison is needed to isolate the contribution of diffusion + guidance from the contribution of better networks and critics.
>
> **A2**: We thank the reviewer for the comment. We would like to clarify that CGPO and methods such as SimBa [1] target different research fields. CGPO focuses on designing better diffusion-based RL (one advanced branch of RL) rather than exploring better policy/critic network architecture, which SimBa does. The two fields are orthogonal rather than competitive. Moreover, CGPO can also be naturally combined with SimBa, for better performance.
>
> Besides, while diffusion-based RL introduces additional computation compared to SAC/TD3, the models used in them remain lightweight. Specifically, the network of diffusion we used in CGPO has only 0.18M, which are close to the model size 0.1M in SAC and TD3. Therefore, we believe our comparison is compute-matched or parameter-matched (especially with other diffusion-based RL algorithms) and the performance gains cannot be attributed to large model capacity, but rather to the proposed diffusion and guidance design.
>
>
> > **Q3**: No sparse-reward simulation benchmarks ...
>
> **A3**: We thank the reviewer for the comment. However, our work does not target sparse-reward RL, which is a separate research direction (e.g., [R4]).  The **exploration** we refer to concerns optimization under highly non-convex reward settings, where insufficient exploration can lead to suboptimal policy. CGPO is designed to address this challenge.
>
> Beisdes, for sparse-reward settings, our method is complementary to existing techniques (e.g., [R4]) and can be readily combined with them.
>
> [R4] Andrychowicz M, Wolski F, Ray A, et al. Hindsight experience replay[J]. Advances in neural information processing systems, 2017, 30.
>
> > **Q4(1)**: In sparse-reward settings, the rectified advantage weights likely stay near zero until the agent first encounters reward, effectively disabling guidance. When rewards are finally observed, would the hard gating in $max(Q̂ − V_\phi, 0)$ create a sharp transition from inactive to active weights?
>
> **A4(1)**: We thank the reviewer for the insightful question. We would like to clarify that, although the immediate rewards are sparse, the learned Q and V functions are not: Q estimates the discounted cumulative reward $Q_{\pi}(s, a) = \mathbb{E}_{\tau \sim \pi} [\sum \gamma^t r_t \mid s, a]$, which propagates reward information to preceding states. As a result, states with higher probability of reaching the goal (or closer to the goal) will naturally have relatively high Q values. Therefore, the weight does not stay uniformly near zero, and the rectified weighting does not induce sharp transition.
>
> > **Q4(2)**: And since the guided actions become supervised regression targets rather than soft policy gradient signals, would this commit the policy more strongly to potentially unreliable critic estimates? Has this scenario been tested, and does the policy overfit to early rewarding experiences?
>
> **A4(2)**: We thank the reviewer for the question. This issue does not arise in our setting. First, we inherit the entropy regularization from QVPO, which ensures persistent exploration during policy optimization and prevents the policy from collapsing to specific early experiences. Therefore, the policy does not get stuck in or overfit to early experiences, which is also supported by our superior performance in mujoco benckmarks.
>
> > **Q5**: What are the values of N (ensemble size), M (quantiles per critic), and k (truncation count)?
>
> **A5**: We uses $N=5$ critics, $M=25$ quantiles per critic, and truncation count $k=2$. We will include these exact values in our final version.
>
>
> > **Q6**: Minor typos: 1) three-fold -> four bullet points, 2) tradd-off -> trade-off, 3) The claim of being "the first success to incorporate diffusion policy into real-world RL"
>
> **A6**: We will correct them or refine our claim in our final version.

---

> > ### Author Rebuttal · Reviewer_8fdp · 2026-04-06
> >
> > Thank you for your response. All my questions have been resolved.

---

> > > ### Author Response · Authors · 2026-04-06
> > >
> > > Thank you for your favorable comments and for acknowledging the revisions detailed in our rebuttal document.

---

### Official Review · Reviewer_nyEK · 2026-03-10

**Soundness:** 3
**Presentation:** 3
**Significance:** 3
**Originality:** 3
**Overall Recommendation:** 4
**Confidence:** 2

**Summary:**

This paper studies online diffusion-based RL and argues that existing diffusion RL methods suffer from an exploration-exploitation mismatch: weighted sampling methods underuse critic information, while gradient-based methods may lose multimodality. To address this, the paper proposes CGPO, which uses training-time critic guidance during the late denoising steps of a diffusion policy to synthesize a single guided action target per state, then trains the actor by weighted denoising regression toward these targets. The method also included a value-calibrated network for stabilizing state weighting and a conservative distributional critic aggregation for more reliable guidance. Experiments on 5 MuJoCo tasks and a Franka robot steup better performance than several RL and diffusion-RL baselines.

**Compliance With Llm Reviewing Policy:**

Affirmed.

**Key Questions For Authors:**

In the actual implementation, is the guidance signal computed as
$\nabla_x [-\hat Q(s, \hat x_0 (x_t, s, t))]$
as Equation 10 suggests,
$\nabla_x \hat Q(s,x)\vert_{x=\hat x_0}$
as Algorithm 2 suggests? If these differ, which one produced the reported results?

Why is SDAC omitted from the main experimental comparison, given that it is discussed in the related work as a closely relevant diffusion-RL baseline? If the issue is code availability or instability, please state that explicitly and, ideally, provide any attempted comparison.

Can you provide direct quantitative evidence for the paper's central mechanism, for example. the evolution of $\nabla_Q(s)$ fromEquation 9 over training, action diversity metrics, or critic-value gap between guided targets and unguided samples? If this evidence is strong across tasks, my assessment would become more positive.

In the real-world setup, how much human intervention and how many fallback demonstrations were actually used for each method? This point matters a lot because it directly affects the validity of the ``from scratch, without demonstrations" claim.

**Limitations:**

This paper partially acknowledges limitations but several important ones remain underexplored. Guided diffusion relies on Q-gradients, which may be highly unreliable early in training. Evaluation focuses on MuJoCo locomotion and two simple robot tasks. Training the actor to imitate guided actions may introduce distribution mismatch.

**Strengths And Weaknesses:**

**Strengths**

The core idea is intuitive and practically motivated. Using critic guidance to generate one refined target action, instead of relying on finite candidate that directly address the bottleneck discussed in Section 4.

The MuJoCo results are strong on the included benchmark set. In Figure 3 and Table 1, CGPO is consistently the top method across all five tasks. The margins are not tiny either, for example on HalfCheetah-v3 CGPO reaches 14368.6 vs. 11385 for QVPO.

Figure 1 gives a helpful high-level picture of the intended trade-off. It makes the intended message legible before getting into the details of Equation 10 to 19.

The paper is explicit in Section 4.2 and again around Algorithm 1 that guidance is used only during training to synthesize targets, while environment interaction and test-time execution use the unguided sampler.  This is a practically sensible system choice, particularly for robotics where control-loop latency matters.

**Weaknesses**

The paper spends a lot of effort arguing that sampling-based policy improvement deteriorates because candidate diversity shrinks and critic separability vanishes, but this is mostly asserted. Figure 2 is a single qualitative t-SNE on one environment, and the paper never measures the critic contrast proxy from Equation 9. The whole method is justified as a targeted fix for that failure mode. Without direct measurements, the narrative is still plausible, but it is not yet scientifically pinned down.

SDAV is discussed in the related work, explicitly as a diffusion-RL method with improved sample efficiency, yet it does not appear in Figure 3 or Table 1. That omission weakens the empirical positioning quite a bit. When a paper claims SOTA among diffusion-based RL methods, leaving out one of the closet recent baselines is a real problem, not a minor citation quibble.

In the main part, Equation 10 defines the guide loss through $\hat{x}_0(x_t, s, t)$, which suggests backpropagating through the denoiser's clean-action prediction. But Algorithm 2 appear to compute the gradient directly with respect to $x = \hat{x}_0$, not with respect to $x_t$. Those are not obviously identical. This is not a notation issue, it changes the actual algorithm. This paper needs to clean this up.

---

> ### Author Rebuttal · Authors · 2026-03-31
>
> We thank Reviewer nyEK for the thoughtful and constructive feedback. We address each point in detail below.
>
> > **Q1**: In the actual implementation, is the guidance signal computed as $\nabla _{x_t}[-\hat{Q}(s,\hat{x} _0(x_t,s,t))]$ as Equation 10 suggests, $\nabla _{x_t} \hat{Q} _\psi(s, x) \big| _{x=\hat{x} _0}$ as Algorithm 2 suggests? If these differ, which one produced the reported results?
>
> **A1**: The guidance signal computed as $\nabla_{x_t}[-\hat{Q}(s,\hat{x}_0(x_t,s,t))]$ as Equation 10. We will correct Algorithm 2 in the revised version. Thank you for pointting the problems.
>
> > **Q2**: Why is SDAC omitted from the main experimental comparison, ...?
>
> **A2**: SDAC is similar to DACER, which is why we chose DACER as a representative method. In addition, we have conducted experiments with SDAC under the same experimental protocol. The results are as follows:
>
> | Task            | SDAC              | CGPO(*)             |
> |-----------------|------------------|--------------------|
> | Ant-v3          | 1404.6 (105.3)   | 7272.1 (74.6)  |
> | HalfCheetah-v3  | 11179.4 (325.1)  | 14368.6 (310.6)|
> | Hopper-v3       | 2957.8 (323.7)   | 4170.5 (165.6)|
> | Humanoid-v3     | 4429.6 (174.2)   | 7131.4 (243.8)|
> | Walker2d-v3     | 4250.8 (411.9)   | 6482.6 (170.9) |
>
> Additional training results are available at this (https://anonymous.4open.science/r/icml-cgpo-C720/results.pdf). We will include these results in the final version to strengthen the empirical comparison.
>
> > **Q3**: The paper spends ... Figure 2 is a single qualitative t-SNE ... paper never measures the critic contrast proxy from Equation 9. ... Without direct measurements ... but it is not yet scientifically pinned down.
>
> > Can you provide direct quantitative evidence for the paper's central mechanism, for example. the evolution of $\Delta_Q(s)$ ...?
>
> **A3**: We conducted further quantitative analysis at (https://anonymous.4open.science/r/icml-cgpo-C720/Ant-v3_sampling_vs_gradient.pdf). During training in the Ant-v3 environment, we **sampled 64 samples** to select the optimal one and calculated the value of $\Delta_Q(s)$ using Equation 9. We additionally calculated $\Delta_Q(s)$ for Q **with and without DSG guidance**. By comparison, we found that the sampling-based method showed a significant increase in $\Delta_Q(s)$ in the early stages, but then **continuously decreased as training progressed**, exhibiting large fluctuations. This precisely validates our qualitative analysis. In contrast, the gradient-based method showed relatively stable $\Delta_Q(s)$ and **achieved a better improvement than the sampling-based method in the later stages**.
>
> > **Q4(1)**: In the real-world setup, how much human intervention and how many fallback demonstrations were actually used for each method?
>
> **A4(1)**: Our 30 Hz closed-loop system begins training and human intervention simultaneously, with the intervention frequency naturally decreasing as the policy matures. CGPO achieves a 0.8 success rate in just 15,784 total steps (approximately 40 minutes), requiring significantly fewer autonomous rollouts than the SAC baseline (17997 steps) to reach a higher performance level. About one-third of the steps in CGPO involve human intervention, and the same is true for SAC.
>
> > **Q4(2)**: This point matters a lot because it directly affects the validity of the "from scratch, without demonstrations" claim.
>
> **A4(2)**: "without demonstrations" here means our method is trained from scratch without any offline pre-training with expert datasets before. Hence, human intervention here cannot be considered to affect the claim of "from scratch, without demonstrations" in the real-world RL.
>
>
> > **Q5(1)**: Guided diffusion relies on Q-gradients, which may be highly unreliable early in training.
>
> **A5(1)**: We have illustrated this problem in **A2** of response to reviewer TDQ5. The unreliability of the early-stage critic is a common issue in RL, and many prior RL methods like TD3 and SAC are affected by it. However, this does not hinder the overall RL training. In fact, even if the policy gradient is incorrect at the early stage, the RL learning mechanism quickly incorporates these samples, allowing the critic to be corrected and the policy to improve over time.
>
>
> > **Q5(2)**: Evaluation focuses on MuJoCo locomotion and two simple robot tasks.
>
> **A5(2)**: We will consider adding more benchmarks for comparison in our final version. However, we clarify that our two real-world robot tasks are challenging for their high-dimensional state (image) and sparse reward settings.
>
> > **Q5(3)**: Training the actor to imitate guided actions may introduce distribution mismatch
>
> **A5(3)**: Regarding to the distribution mismatching, we have illustrated in **A1** of response to reviewer TDQ5. Training the actor to imitate guided actions are actually equivalent to the RL objective as analyzed in Appendix A. Hence, there is no distribution mismatch problem.

---

### Official Review · Reviewer_FS8k · 2026-03-11

**Soundness:** 4
**Presentation:** 2
**Significance:** 3
**Originality:** 3
**Overall Recommendation:** 5
**Confidence:** 4

**Summary:**

This paper presents the RL methods that can conduct exploration while utilizing the exploitation policy. With guidance of the critic, the diffusion policy can be converged more stably by making the policy stay nearby the exploitation-done state spaces. This approach could solve both problems in gradient-based methods and weight-based methods. Moreover, the impressive point was that they already designed the method based on the assumption of the instability or incorrectness of q-value. The authors evaluated their method on both simulation and real-world environment and demonstrateed their method's performance.

**Compliance With Llm Reviewing Policy:**

Affirmed.

**Key Questions For Authors:**

Q1. Can you explain more details about Figure 2. How can we figure out that the candidate diversity shrinks and also within-set separability decreases? It is not that I don't understand the caption but I want to know what is related in the pictures (tsne plots).

Q2. Can you measure operation time? How much Hz your closed-loop system shows?

**Limitations:**

Several hyperparameters in their method may affect the policy performance and figuring out the correct value could be harsh.

**Strengths And Weaknesses:**

Strength
- Strong problem definition: They define the basic problems in diffusion policy-based RL.
- Well-formulated method: It was impressive that the authors have already thought and prepared about the case of q-value incorrectness. Also, they supplied the abundant appendix to help understand their main texts.
- Harsh evaluation: By evaluating their methods in both real-world and simulation, they proved their methods sufficiently.


Weakness
- In Figure 2, this reviewer can not nofity what the exact problem is.

---

> ### Author Rebuttal · Authors · 2026-03-31
>
> We thank reviewer FS8k for their recognition of our work. In the following, we will address his/her problems point by point.
>
> > **Q1**: Can you explain more details about Figure 2. How can we figure out that the candidate diversity shrinks and also within-set separability decreases? It is not that I don't understand the caption but I want to know what is related in the pictures (tsne plots).
>
> **A1**: Thanks for your problem. In Figure 2, we aim to illustrate that, as training progresses, it becomes increasingly difficult for sampling-based diffusion RL methods like QVPO to sample a high-quality action candidate for the diffusion loss. That leads to slow policy improvement in the later stages of training. In that case, we project action samples generated by the diffusion policy at different training stages into 2D using t-SNE, and classify them into high-quality (green) and low-quality (red) actions based on Eq. (9). As training progresses (from left to right), the number of high-quality actions clearly decreases. This supports our claim that sampling-driven methods struggle to obtain high-quality action samples in the later stages of training.
>
> Moreover, for better visualization, we also provide a more detailed Figure 2 in (https://anonymous.4open.science/r/icml-cgpo-C720/tsne.pdf). We hope this new figure will help you understand the limitation of sampling-driven methods and we will add it in the final version.
>
> > **Q2**: Can you measure operation time? How much Hz your closed-loop system shows?
>
> **A2**: Our closed-loop system operates at a control frequency of approximately 30 Hz. During the training process, human intervention begins immediately at the initial stage with a higher frequency, which gradually decreases as the policy matures. The entire training process converges within 15,784 total steps, requiring 41 minutes of physical interaction time.

---

> > ### Author Rebuttal · Reviewer_FS8k · 2026-04-03
> >
> > Thank you for your response. I will consider adjusting my score : )

---

> > > ### Author Response · Authors · 2026-04-05
> > >
> > > Thank you for your positive feedback and for acknowledging the revisions made in the rebuttal.

---

### Official Review · Reviewer_TDQ5 · 2026-03-11

**Soundness:** 2
**Presentation:** 3
**Significance:** 2
**Originality:** 2
**Overall Recommendation:** 3
**Confidence:** 3

**Summary:**

This paper proposes CGPO (Critic-Guided Diffusion Policy Optimization). The authors address a conflict in existing diffusion-based RL methods between early-stage exploration and late-stage refinement: one class relies on candidate action sampling, but candidate separability decreases later; the other directly uses Q-gradient, which can sacrifice policy diversity. CGPO introduces critic-guided, training-free guidance during the denoising process of the diffusion actor to generate a guided target action, then updates the actor via weighted denoising regression and entropy regularization. Stabilization mechanisms such as truncated quantiles, late-step DSG guidance, and a value-calibrated network are also incorporated. Experiments include five MuJoCo locomotion tasks and the peg-in-hole task on a Franka robotic arm.

**Compliance With Llm Reviewing Policy:**

Affirmed.

**Final Justification:**

The author's rebuttal did not address my concerns. I will maintain the current score.

**Key Questions For Authors:**

1. Can the authors prove the relationship between guided-target denoising regression and the RL objective?
2. Regarding the unreliability of the early-stage critic, is there evidence that truncated quantiles, late-step guidance, and value calibration are sufficient to maintain stability?
3. Why does the real-robot experiment compare only with SAC? Could comparisons with stronger diffusion-policy or real-world RL baselines be added?
4. Can mechanism-specific metrics over training (e.g., critic ranking reliability, candidate separability, guided target quality) be included to more directly support the method’s motivation?

**Limitations:**

CGPO is primarily a combination of mechanisms rather than a theoretical innovation; ablation depth is insufficient to support strong claims for real-world diffusion RL.

**Strengths And Weaknesses:**

**Strengths**

- Clear motivation. As the policy converges, candidate actions become similar, and the critic’s ranking signal weakens, leading to self-imitation. CGPO is proposed specifically to address this issue.
- Beyond introducing critic-guided refinement, CGPO designs stabilization mechanisms including truncated quantiles, late-step guidance, DSG, and value-calibrated weighting, forming a systematic approach.
- Effective experimental results. CGPO achieves the best overall performance on MuJoCo tasks with stable learning curves; in the real-robot peg-in-hole task, it outperforms SAC.

**Weaknesses**

- Insufficient theoretical support. The consistency between guided-target denoising regression and the final RL objective is not proven. The actor may only learn to fit the guidance-corrected target rather than directly optimizing the RL objective.
- Stability relies on empirical methods. The early-stage critic is the least reliable, and the paper only uses empirical methods to mitigate this, without analyzing whether the method degrades or misleads the actor under critic distortion.
- Engineering-focused innovation. The method is a combination of multiple mechanisms, lacking theoretical novelty or rigorous analysis.
- Insufficient ablation and analysis. The contribution of each component to method effectiveness is not fully shown, and key metrics are missing (e.g., critic ranking reliability, candidate separability, guided target quality).

---

> ### Author Rebuttal · Authors · 2026-03-31
>
> We would like to thank the Reviewer TDQ5 for the detailed and constructive comments. In the following, we have provided an item-by-item response to the comments.
>
> > **Q1**: Insufficient theoretical support and engineering-focused innovation. CGPO is a combination of mechanisms rather than a theoretical innovation. Can you prove the relationship between CGPO and the RL objective?
>
> **A1**: We clarify that CGPO is derived from a formal theoretical framework, not a combination of mechanisms. According to the RL objective in (Appendix. A.1), we can derive the optimal policy
>
> $\pi^*(a|s) \propto \pi_{old}(a|s) \exp(\frac{1}{\beta} Q(s,a)),$
>
> converting the RL task into a inference problem. In (Appendix. A.2-A.3) we prove that our diffusion policy update guarantees policy improvement. In (Appendix. A.4-A.5), we provide a rigorous proof that our critic-guided denoising process is the exact implementation of performing inference from $\pi^*$.
>
> > **Q2**: Stability relies on empirical methods. Regarding the unreliability of the early-stage critic, is there evidence that truncated quantiles, late-step guidance, and value calibration are sufficient to maintain stability?
>
> **A2**:  Our words may mislead you. In fact, the unreliability of early-stage critic is a common issue in RL, and many prior RL methods like TD3 and SAC are affected by it. Distributional critic like truncated quantiles [R1] are developed to address this problem and their effectiveness has been verified in the previous work.
>
> Besides, We clarify that last-step guidance is designed for **avoiding out-of-distribution critic queries** and value calibration is developed for **more accurate weight** rather than the unreliability of the early-stage critic. Hence, **they serve to improve performance of CGPO, instead of being strictly necessary for stablizing the training in CGPO**. To further address your concern, we add more ablation studies on truncated quantiles (https://anonymous.4open.science/r/icml-cgpo-C720/Ant_cgpo_vs_without_tqc.pdf), late-step guidance (as shown in Figure 6(b)), and value calibration (https://anonymous.4open.science/r/icml-cgpo-C720/Ant_cgpo_vs_without_v_net.pdf): with these components, CGPO can achieve better performance with low variance in reward.
>
> [R1] Kuznetsov A, Shvechikov P, Grishin A, et al. Controlling overestimation bias with truncated mixture of continuous distributional quantile critics[C]//International conference on machine learning. PMLR, 2020: 5556-5566.
>
> > **Q3**: Why does the real-robot experiment compare only with SAC? Could comparisons with stronger diffusion-policy or real-world RL baselines be added?
>
> **A3**: We clarify that we actually compare against HIL-SERL [R2], a recent real-world RL method that uses SAC as its RL algorithm, rather than vanilla SAC alone. Besides, comparisons with stronger diffusion-based RL would be valuable. However, no prior diffusion RL methods have been applied to real-world RL. Our work is **the first diffusion-based RL work** in this field, and SAC-based methods remain the strongest available baselines in this setting.
>
> [R2] Luo J, Xu C, Wu J, et al. Precise and dexterous robotic manipulation via human-in-the-loop reinforcement learning[J]. Science Robotics, 2025, 10(105): eads5033.
>
> > **Q4**: Insufficient ablation and analysis...(e.g., critic ranking reliability, candidate separability, guided target quality) be included to support the method’s motivation?
>
> **A4**: (1) The effectiveness of critic ranking reliability has been verified in [R1] (2) Regarding candidate separability and guided target quality, we conduct an extra experiment on $\Delta_Q$ in CGPO and sampling-driven method (https://anonymous.4open.science/r/icml-cgpo-C720/Ant-v3_sampling_vs_gradient.pdf: as training progresses, the $\Delta_Q$ of sampling-based methods decreases (i.e., it is hard to obtain high-quality actions in the later training stages), whereas CGPO consistently maintains a certain $\Delta_Q$ (i.e., stable improvement in rewards). This indicates that the guided action targets in CGPO are of higher quality. Besides, optimizing with guided action target has been proved to be equivalent to RL objective. In that case, both theoretical and empirical evidence can support our motivation.
>
> We conducted more ablation studies at (https://anonymous.4open.science/r/icml-cgpo-C720/). Specifically, we evaluate:(1) Remove the $V_{\phi}$ network. (2) Remove the double q network (DDQN). (3) Do not perform guidance, perform naive guidance, and perform DSG guidance. (4) Replace the Truncated Quantiles Critic with the standard Critic. To avoid randomness in the experiment, we ran three seeds in each of the Ant-v3 and Walker2d-v3 environments.

---

> > ### Author Rebuttal · Reviewer_TDQ5 · 2026-04-02
> >
> > Thank you for the detailed rebuttal. The added clarifications and ablations are helpful. I have carefully read Appendix A, but I still have several important questions that are central to assessing the paper.
> >
> > 1. **On Appendix A.5.**
> >    Appendix A.5 is presented as supporting an “exact implementation of inference from $\pi^*$,” but the transitions in Eqs. (45)–(48) appear to involve nontrivial approximations. Could the authors clarify whether this part is intended as a heuristic derivation or a strict proof? If it is meant to be a strict proof, please state the required assumptions and, if possible, provide a formal derivation or an error bound for these steps.
> >
> > 2. **On the link between the theoretical objective and the actor update.**
> >    The paper derives a KL-regularized RL objective in Appendix A.1–A.3, but it is still unclear why optimizing the guided-target denoising losses in Eqs. (18)–(19) is equivalent to optimizing that objective. Could the authors clearly explain the missing link between inference from $\pi^*$ and the guided-target denoising regression used to train the actor?
> >
> > 3. **On the real-world cold-start setting.**
> >    The real-world setting is still unclear. The main text and conclusion emphasize “from scratch / without demonstrations,” but Section 5.2 and Appendix B.1 indicate the use of initial expert demonstrations, corrective interventions, and fallback to full demonstrations. Could the authors clarify the exact real-world training protocol, including:
> >    * how many initial demonstrations are used,
> >    * how often corrective interventions occur,
> >    * under what conditions the fallback to full demonstrations is triggered,
> >    * whether these demonstration/intervention data are added to the replay buffer and used for policy updates,
> >    * and whether any warm-start or pretrained initialization is used beyond the pretrained visual encoder?
> >
> > These points are important for my evaluation. Can you provide the code for the core algorithm module of the simulation experiment, especially the part related to cold-start handling? Providing the code would make the cold-start setting and the corresponding claims easier to verify.

---

> > > ### Author Response · Authors · 2026-04-05
> > >
> > > Thank you for your response. We have provided an item-by-item response to your new comments.
> > >
> > > > **Q1**: On Appendix A.5. Appendix A.5 is presented as supporting an “exact implementation of inference from,” but the transitions in Eqs. (45)–(48) appear to involve nontrivial approximations. Could the authors clarify whether this part is intended as a heuristic derivation or a strict proof? If it is meant to be a strict proof, please state the required assumptions and, if possible, provide a formal derivation or an error bound for these steps.
> > >
> > > **A1**: The words "exact guided score" may mislead you. Eqs. (45)–(48) are actually approximations according to Jensen's gap in Appendix A.4. However, this approximation is **commonly applied in diffusion guidance** [R6, Eq.15], [R7, Eq.8]. We will correct the words in our final version.
> > >
> > > [R6] Chung H, Kim J, Mccann M T, et al. Diffusion posterior sampling for general noisy inverse problems[J]. arXiv preprint arXiv:2209.14687, 2022.
> > >
> > > [R7] Yang L, Ding S, Cai Y, et al. Guidance with spherical Gaussian constraint for conditional diffusion[J]. arXiv preprint arXiv:2402.03201, 2024.
> > >
> > > > **Q2**: On the link between the theoretical objective and the actor update. The paper derives a KL-regularized RL objective in Appendix A.1–A.3, but it is still unclear why optimizing the guided-target denoising losses in Eqs. (18)–(19) is equivalent to optimizing that objective. Could the authors clearly explain the missing link between inference from $\pi^{*}$ and the guided-target denoising regression used to train the actor?
> > >
> > > **A2**: Thank you for the question. The inference process of the guided diffusion corresponds to the optimal policy $\pi^{k+1}$ in Eq. (23) using the approach detailed in Eq. (48), which holds the score of $\pi^{k+1}$: $\nabla_a\log\pi_{k+1}(a|s)=\nabla_a\log\pi_k(a\mid s)+\frac{1}{\eta}\nabla_aQ (s,a)$. In that case, the optimal solution of the guided-target denoising regression in Eqs. (18)–(19) is actually $\pi^{k+1}$ in Eq. (23). Notably, the weights in loss are just a technique applied for balance between states, and do not affect the optimality of policy theoretically.
> > >
> > > > **Q3(1)**: On the real-world cold-start setting. The real-world setting is still unclear. The main text and conclusion emphasize “from scratch / without demonstrations,” but Section 5.2 and Appendix B.1 indicate the use of initial expert demonstrations, corrective interventions, and fallback to full demonstrations.
> > >
> > > **A3(1)**：Thank you for your questions. Within the HILSERL framework, we use 5 initial expert data trajectories, and these demonstrations serve solely to improve the sample efficiency of online learning. We would like to clarify that “without demonstrations/from scratch” aims at illustrating that our method is trained entirely without any offline pre-training. We will modify our words in the final version.
> > >
> > > > **Q3(2)**: Could the authors clarify the exact real-world training protocol,
> > > > 1. including: how many initial demonstrations are used,
> > > > 2. how often corrective interventions occur,
> > > > 3. under what conditions the fallback to full demonstrations is triggered,
> > > > 4. whether these demonstration/intervention data are added to the replay buffer and used for policy updates, and
> > > > 5. whether any warm-start or pretrained initialization is used beyond the pretrained visual encoder? These points are important for my evaluation.
> > >
> > > **A3(2)**:
> > > 1. We use 5 initial demonstrations.
> > > 2. CGPO was trained for a total of 15,784 steps, with 5393 steps consisting of 4610 corrective interventions and 783 full demonstrations.
> > > 3. We trigger a full demonstration/partial intervention after 3-5 consecutive obvious/partial failures operated by the diffusion policy itself.
> > > 4. Yes, the data trajectory of the intervention was used for policy updates.
> > > 5. No, our diffusion policy didn't use any warm-start or pretrained initialization except for a pre-trained visual encoder ResNet10.
> > >
> > > > **Q3(3)**: Can you provide the code for the core algorithm module of the simulation experiment, especially the part related to cold-start handling? Providing the code would make the cold-start setting and the corresponding claims easier to verify.
> > >
> > > **A3(3)**: We've provided the cold-start code of CGPO in real-world tasks at (https://anonymous.4open.science/r/icml-cgpo-C720/cgpo_cold_start.py) and the cold-start code of CGPO in simulation tasks at (https://anonymous.4open.science/r/icml-cgpo-C720/cgpo_simulation.py).

---

### Decision · Program_Chairs · 2026-04-30

**Decision:**

Accept (regular)

**Comment:**

In this paper, the authors proposed a new practical way of optimizing on-line diffusion policies, combining many existing tools. While previous methods are based on multiple sampling from the current policy and reweighted noise model regression, which requires increasing sampling rates, the proposed method obtains a sample of high value through training-free Q guidance, and use this sample’s weight for reweighted noise model regression. Although additional state value function learning is required due to lack of multiple samples which automatically yields an state value estimate by averaging Q values over multiple actions, the overall algorithm provides an efficient way to improve diffusion policies. It is recommended that in the final version authors incorporate the reviewers’ comments, in particular, thorough ablation study and analysis (e.g., sphere hardening is valid for hundreds or thousands of dimensions. It is questionable indeed this normalization matters for tens of dimensions), key metric analysis, recent baselines, and revise the paper.